# Tail state limited photocurrent collection of thick photoactive layers in organic solar cells

Jiaying Wu[1], Joel Luke [2], Harrison Ka Hin Lee[3], Pabitra Shakya Tuladhar[1], Hyojung Cha [1], Soo-Young Jang[1,4], Wing Chung Tsoi[3], Martin Heeney [1], Hongkyu Kang [1,4]*, Kwanghee Lee[4], Thomas Kirchartz [5,6]*, Ji-Seon Kim [2]* & James R. Durrant [1,3]*

We analyse organic solar cells with four different photoactive blends exhibiting differing dependencies of short-circuit current upon photoactive layer thickness. These blends and devices are analysed by transient optoelectronic techniques of carrier kinetics and densities, air photoemission spectroscopy of material energetics, Kelvin probe measurements of work function, Mott-Schottky analyses of apparent doping density and by device modelling. We conclude that, for the device series studied, the photocurrent loss with thick active layers is primarily associated with the accumulation of photo-generated charge carriers in intra-bandgap tail states. This charge accumulation screens the device internal electrical field, preventing efficient charge collection. Purification of one studied donor polymer is observed to reduce tail state distribution and density and increase the maximal photoactive thickness for efficient operation. Our work suggests that selecting organic photoactive layers with a narrow distribution of tail states is a key requirement for the fabrication of efficient, high photocurrent, thick organic solar cells.

[1] Department of Chemistry and Centre for Plastic Electronics, Imperial College London, London W12 0BZ, UK. [2] Department of Physics and Centre for Plastic Electronics, Imperial College London, London SW7 2AZ, UK. [3] SPECIFIC, College of Engineering, Bay Campus, Swansea University, Swansea SA1 8EN, UK. [4] Research Institute for Solar and Sustainable Energies, Gwangju Institute of Science and Technology, Gwangju 61005, Republic of Korea. [5] IEK5-Photovoltaik, Forschungszentrum Jülich, 52425 Jülich, Germany. [6] Faculty of Engineering and CENIDE, University of Duisburg-Essen, Carl-Benz-Strasse 199, 47057 Duisburg, Germany. *email: gemk@gist.ac.kr; t.kirchartz@fz-juelich.de; ji-seon.kim@imperial.ac.uk; j.durrant@imperial.ac.uk

O ver the last decade, there has been substantial progress in the development of organic photovoltaic (OPV) devices, driven by advances in molecular design, material processing, and device engineering. Power conversion efficiencies for the state of art laboratory-scale devices now exceed 16% for single junction and 17% for tandem cells[1,2], enabled in particular by the recent development of non-fullerene acceptors[3], and promising long-term stabilities have been reported for devices with efficiencies over 10%[4,5]. This progress is increasing the commercial potential of OPVs. However, to enable large scale processing of OPV devices, one of the key challenges is to achieve high device efficiencies with photoactive layers thick enough for scalable printing methodologies, which typically require thicknesses more than 300 nm. For laboratory-scale spin-coated devices, optimum OPV performance is typically obtained with photoactive layer thicknesses less than 100 nm, resulting from the compromise between strong light absorption and efficient charge collection. Only a few material systems, including most notably the lower efficiency P3HT:$PC_{61}BM$ system[6,7], have been shown to work effectively with photoactive layer larger than 300 nm[8–10]. This thickness limitation for OPV has typically been associated with non-geminate charge recombination losses during charge transport to the device electrodes, often quantified regarding the charge carrier mobility-lifetime ($\mu\tau$) product[11], or analogous combinations of the bimolecular recombination coefficient and the mobility[12]. Larger absorber thicknesses lead to increased charge collection losses due to non-geminate recombination, resulting in low device FF and consequently lower device performance[13,14]. Above a certain material dependent thickness threshold, thicker devices also typically show lower short-circuit currents $J_{sc}$, despite an increase in the number of absorbed photons[15]. In the study herein, we demonstrate that for the organic solar cells studied the reduction in $J_{sc}$ with thicker absorber layer does not result from an insufficiently high mobility-lifetime product, but rather from charge accumulation in absorber layer tail states. This charge accumulation screens the device electric field from most of the absorber layer, thereby preventing efficient charge collection from outside the relatively thin space-charge layer.

After optical excitation of organic donor/acceptor blends, the resultant charge carriers must be selectively extracted from the absorber layer. For organic semiconductors, the charge carrier diffusion lengths are typically thought to be relatively small (16 nm for typical values for $\mu\tau$ of $10^{-10}$ cm$^2$ V$^{-1}$ as reported in the literature)[11,15,16], such that purely diffusive transport does not lead to efficient charge extraction (CE). To address this limitation, organic solar cells typically comprise a thin photoactive layer sandwiched between two electrodes with different work functions. This results in an internal electric field across the device at short circuit, which enables faster, and therefore more efficient, extraction of photogenerated electrons and holes. As such, obtaining a high value for drift length $L_{dr}$ versus photoactive layer thickness $d$, and therefore efficient charge collection for thick devices, requires a high value for the mobility-lifetime product, with several studies in the literature addressing this issue[15,17,18].

The standard field-driven charge collection model and mobility-lifetime product analysis described in the preceding paragraph is based upon several assumptions which limits its practical application to predicting OPV device performance[11,16,17]. Of particular relevance to the study herein, it assumes that the photoactive layer is fully depleted, such that the device built-in electric field enables efficient drift driven charge collection across the whole photoactive layer (i.e., it assumes an absence of any space charge which may screen such fields). Studies to date challenging this assumption, and its relevance to the difficulty of achieving efficient OPV device performance with

relatively thick photoactive layers, have focused primarily upon the impact of unintentional doping species in the photoactive layer[19,20], or asymmetric mobilities in screening the device built-in electric field[15]. Such screening has been shown to result in poor photocurrent collection from photons absorbed in the low electric field region[15,21,22], and the appearance of a 'space-charge voltage penalty', particularly caused by imbalanced mobilities, limiting device performance[23,24], emphasising the importance of achieving low doping densities and matched mobilities for efficient, thick OPV devices.

Intra-bandgap tail states have been widely observed in disordered materials, including organic semiconductors. In organic semiconductors, local structural, intra- or intermolecular disorder or chemical defects can result in exponential tails of shallow trap states extending from the conduction band and valence band ($C_B$/$V_B$) edges into the bandgap[25,26]. The density and energetic distribution of these tail states depend on the material morphology and energetics. In organic donor:acceptor blend photovoltaic devices, such tail states have been extensively characterised, with charge densities measured to accumulate in these states of up to from $10^{15}$ to $10^{16}$ cm$^{-3}$ under short-circuit conditions[27], and from $10^{16}$ to $10^{17}$ cm$^{-3}$ at open circuit[28,29]. Research to date has primarily focused on the impact of these tail states on the charge carrier mobilities and recombination kinetics in devices[30–33], but has not considered in detail the potential impact of such tail states on space-charge effects.

As introduced above, studies addressing the challenge of achieving efficient OPV devices with thick photoactive layers have focused on the need to increase the mobility-lifetime product, and the impact of this on enhancing the fill factor of thick devices[11,15,34,35]. In this study herein, we address a separate, and previously largely undiscussed, challenges for the fabrication of thick efficient OPV devices—the need to avoid excessive charge accumulation in intra-bandgap tail states which can screen the internal field in the device, and thereby reduce the efficiency of photocurrent generation.

## Results and discussion

**Thickness-dependent device performance.** In the study herein, we employed photoactive blends of three different organic donors with the acceptor $PC_{71}BM$, namely the small molecule donor BTR[36], and the polymer donors PCDTBT and DT-PDPP2T-TT[8,37]. For the low bandgap polymer DT-PDPP2T-TT, devices were also fabricated after the polymer had been purified by Soxhlet extraction (purified DT-PDPP2T-TT). This extraction process was employed to remove lower molecular weight polymer fractions. It has been suggested that a high fraction of larger molecular weight material is more favourable for thick device performance[38,39]. These blends all showed more than 6% PCE at an optimum thickness in typical organic bulk heterojunction (BHJ) device architectures (see Supplementary Fig. 1).

Figure 1 shows normalised device parameters measured under simulated AM 1.5G illumination for the four blend systems as a function of photoactive layer thickness, the original current–voltage ($J$–$V$) response curves can be seen in Supplementary Fig. 2. We note that the device open-circuit voltage $V_{oc}$ showed only minor thickness dependence and is therefore not discussed further. As can be seen from Fig. 1, the efficiency of PCDTBT devices drops off rapidly when the photoactive layer thickness is increased over 80 nm due to a loss of both $J_{sc}$ and FF, typical of the thickness dependence data observed for many OPV blend systems. In contrast, the DT-PDPP2T-TT, BTR and purified DT-PDPP2T-TT blends all showed much less severe thickness dependencies, with efficient device performance being maintained up to a maximum photoactive layer thickness $L_{max}$ of

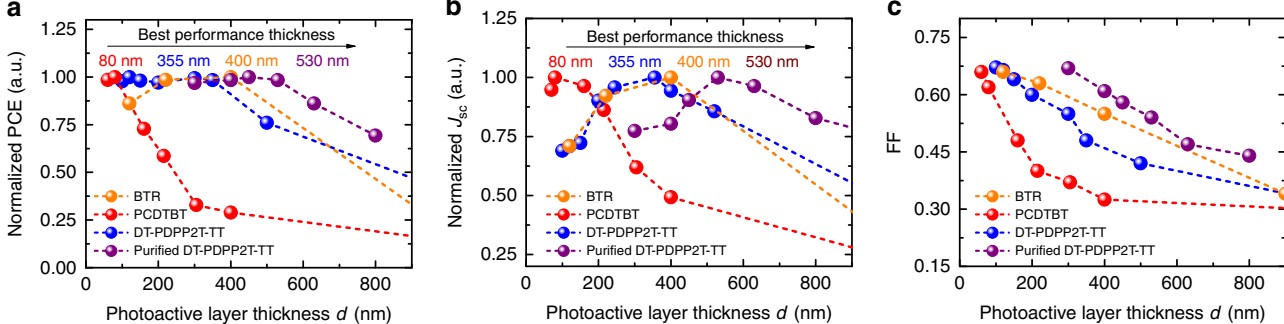

**Fig. 1** Thickness-dependent device performance. PCDTBT:PC$_{71}$BM, original DT-PDPP2T-TT:PC$_{71}$BM, BTR:PC$_{71}$BM and purified DT-PDPP2T-TT:PC$_{71}$BM are compared with thickness-dependent device performance. **a** Normalised PCE as a function of photoactive layer thickness, labelled thickness is assigned the best performing thickness. **b** Normalised $J_{sc}$ as a function of photoactive layer thickness, labelled thickness is associated with the best performing thickness. **c** Normalised FFs as a function of photoactive layer thickness

355 nm for DT-PDPP2T-TT as donor polymer, 400 nm for BTR and 530 nm for purified DT-PDPP2T-TT. All devices show a drop off in FF with increasing device thickness, with this drop off being most severe for PCDTBT and least severe for purified DT-PDPP2T-TT. As discussed previously in the literature[40–42], the drop off in FF can be understood in terms of increased bimolecular recombination losses during charge transport. Both devices show near Langevin behaviour observed in Supplementary Fig. 3, in agreement with their thickness dependence of FF. We also observed there were no obvious changes in the film roughness for both PCDTBT:PC$_{71}$BM and DT-PDPP2T-TT:PC$_{71}$BM blended films with different thicknesses (see Supplementary Fig. 4). It is striking that there is an excellent correlation between the optimum thickness for $J_{sc}$ and efficiency, stressing the importance of understanding this dependence. The values of $J_{sc}$ show an optimum at a finite photoactive layer thickness. At low device thicknesses, $J_{sc}$ increases with increased photoactive layer thickness, which we attribute to enhanced light absorption; when combined with modest losses in FF, this results in essentially thickness independent efficiencies up to $L_{max}$ (the optimal thickness) for DT-PDPP2T-TT, BTR and purified DT-PDPP2T-TT. Our observation of $J_{sc}$ increasing with active layer thickness but FF decreasing has also been reported in the literature[43], and clearly indicates different dependencies of $J_{sc}$ and FF upon active layer thickness. At thicknesses above $L_{max}$, $J_{sc}$ and efficiency drop for reasons that will be investigated in more detail herein.

**Bimolecular recombination losses at short circuit**. The dependence of $J_{sc}$ on photoactive layer thickness $d$ and illumination intensity $\Phi$ can be expressed most simply by a model in which non-geminate (bimolecular) recombination during charge transport is the main loss pathway, and assuming charge photogeneration to be field independent $J_{sc}(\Phi, d) = J_{max}(\Phi, d) - J_{NGR}(\Phi, d)$. The short-circuit current density $J_{sc}$ is thus determined by the difference between the charge maximum photogeneration flux $J_{max}$ and non-geminate recombination flux $J_{NGR}$. $J_{max}$ can be determined experimentally from corrected photocurrent data measured at a sufficiently large reverse bias to ensure efficient field-driven charge collection. The measurement results of corrected photocurrents as a function of photoactive layer thicknesses are shown for PCDTBT and DT-PDPP2T-TT blends in Supplementary Fig. 5. The values of $J_{max}$ as a function of photoactive layer thickness are compared with measured values for short-circuit photocurrent $J_{sc}$. The increase of $J_{max}$ with increasing photoactive layer thickness observed for both blends can be assigned to increased light absorption with thicker absorber layers, modulated by optical interference effects within the devices, as has been widely analysed elsewhere[13,14,44]. It is apparent

that as the photoactive layer thickness is increased above 100 nm, PCDTBT devices show an increasing deviation between $J_{max}$ and $J_{sc}$, indicative of increasing charge carrier losses at short circuit. In contrast for the DT-PDPP2T-TT devices, $J_{max}$ and $J_{sc}$ only deviate for device thicknesses above 300 nm, indicative of significantly reduced thickness-dependent charge carrier losses for this blend.

We now turn to quantify the magnitude of the non-geminate recombination flux $J_{NGR}$ at short circuit for the PCDTBT and DT-PDPP2T-TT device series, in order to determine whether this recombination flux is sufficient to explain the deviations between $J_{max}$ and $J_{sc}$ apparent for the thicker devices in Fig. 2. $J_{NGR}$ at short circuit can be quantified by determining $J_{NGR}$ at open circuit as a function of charge density by CE and transient photovoltage (TPV) techniques, and then using CE measurements at short circuit to determine the charge density, and thereby corresponding recombination flux $J_{NGR}$ at short circuit[29,45–47]. Alternatively, $J_{NGR}$ can be determined from the light intensity dependence of $J_{sc}$ (see Supplementary Fig. 6), where a deviation of $J_{sc}$ versus light intensity from linearity can be assigned to increasing non-geminate recombination losses at higher light fluxes[35]. Both approaches were employed for the PCDTBT and DT-PDPP2T-TT blend devices studied herein to determine the magnitude of $J_{NGR}(d)$ at short circuit under one sun irradiation. The resultant plots of $(J_{sc}(d) + J_{NGR}(d))$ are also shown in Fig. 2 employing either the CE/TPV analysis (orange line) or linearity analysis (pink line). It is apparent that the correction of $J_{max}$ for non-geminate recombination losses is not sufficient to explain the deviation between $J_{max}$ and the measured $J_{sc}$ for thicker devices with either blend. For example, both analyses indicate a value of the non-geminate recombination flux at short circuit under one sun irradiation for the 400 nm thick PCDTBT device of 1.8 mA cm$^{-2}$ (calculations are described in detail showed Supplementary Note 1). However, the measured deviation between $J_{max}$ and $J_{sc}$ for this device under one sun is around 9 mA cm$^{-2}$, i.e. five times larger than the non-geminate recombination losses determined by either analysis. It can thus be concluded that the previously reported methodologies for determining non-geminate recombination losses in OPV devices at the short circuit are unable to explain the loss of photocurrent observed for both device series for thick photoactive layers.

Before analysing the origin of this discrepancy, we note that the CE and TPV analyses are able to account for the more severe reduction in FF with device thickness for the PCDTBT devices than for the DT-PDPP2T-TT devices. These analyses yield one sun mobility-lifetime ($\mu\tau$) products of $1.6 \times 10^{-11}$ and $1.4 \times 10^{-9}$ cm$^2$ V$^{-1}$ for thin PCDTBT and DT-PDPP2T-TT devices near the maximum power point operation charge carrier density, respectively (see Supplementary Fig. 7), with the lower $\mu\tau$

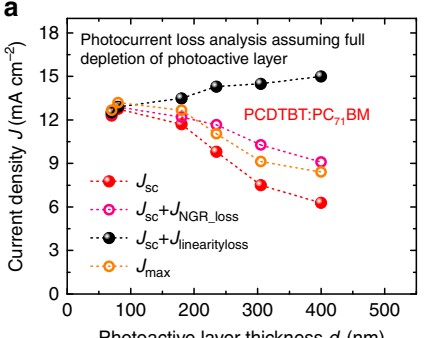
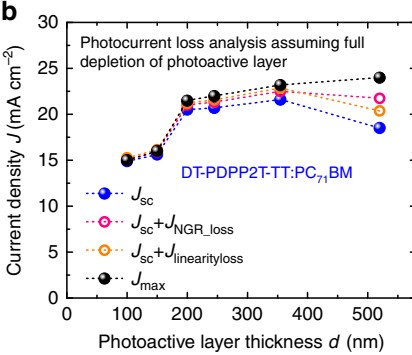

**Fig. 2** Photocurrent analysis at short-circuit condition. **a** Photocurrents of PCDTBT:PC$_{71}$BM are plotted device as a function of photoactive layer thickness: red balls are $J_{sc}$ measured under one sun solar simulator; black balls are $J_{max}$ measured photocurrent at reverse bias under on sun condition which represents the maximum photocurrent that can be generated at short circuit without any loss; orange and pink open circles are photocurrents added back the non-geminate loss and bimolecular linearity loss, respectively, both analyses relates to the consideration of the loss of photocurrent origins from bimolecular recombination. **b** Photocurrents of DT-PDPP2T-TT:PC$_{71}$BM are plotted as a function of photoactive layer thickness. Similarly, the blue balls show the measured short-circuit currents, the black balls indicate the maximum photocurrent can be generated, the orange and pink open circles are the calculated photocurrents without measured bimolecular recombination loss

product obtained for PCDTBT being consistent with the more severe drop off of FF with photoactive layer thickness. We also note that these analyses were able to correctly reproduce the J–V curves of thin PCDTBT and DT-PDPP2T-TT devices assuming the photogeneration current density equals to $J_{sc}$ (See Supplementary Fig. 8), confirming the validity of these analyses for thin devices and indicating that for both devices charge generation is field independent (i.e., an absence of field-dependent geminate recombination losses), and thus ruling such effects out as possible origins of the deviation between $J_{sc}$ and $J_{max}$ apparent from Fig. 2. Similarly, analyses of the $\mu\tau$ products (based on Hecht equation approximation) also indicate that this parameter cannot explain the thickness dependence of $J_{sc}$ (see Supplementary Fig. 9 and Supplementary Note 3)[17,48].

**Determination of depletion layer thickness.** It has previously been suggested that photocurrent generation for thick organic solar cells may be limited if the photoactive layer is not fully depleted due to the space-charge (depletion) layer width being smaller than the device thickness[19,20,23,24]. This discrepancy between depletion width $w$ and photoactive layer thickness $d$ can lead to substantial losses in photocurrent due to an inefficient collection of charges generated in the low field region outside the high field depletion region. We employed two techniques to assay the depletion layer width for the devices studied herein. The first technique is based on TPV/CE analyses of charge carrier lifetimes as a function of charge density for different photoactive layer thicknesses[19]. Assuming only charges in the depletion region contribute to such effective recombination kinetics data (see examples in Supplementary Fig. 10), these data allow us to determine the effective width of the photoactive layer, corresponding to whichever is smaller of either $w$ or $d$. From this analysis, as shown in Fig. 3, we obtain values for the depletion width of PCDTBT and DT-PDPP2T-TT devices of 80 nm and 400 nm, respectively. Alternatively, $w$ can be determined from thickness-dependent capacitance measurements obtained from small perturbation photoexcitation differential capacitance data as plotted in Supplementary Fig. 11. This analysis indicates that for all polymer devices studied, except DT-PDPP2T-TT devices, the measured capacitances deviate from this ideal behaviour for photoactive layer thicknesses above 100–200 nm. Instead, they tend to a constant value at high photoactive layer thicknesses, consistent with space-charge layer limitations. In contrast, for the DT-PDPP2T-TT devices, an ideal behaviour is observed up to

300 nm. These capacitance data therefore also indicate a depletion layer width of around 100 nm for PCDTBT devices, but more than 300 nm for DT-PDPP2T-TT devices, in agreement with our TPV/CE analysis. This small effective space-charge width of PCDTBT device can explain the failure of the bimolecular loss analysis detailed above, which is based on the assumption of the electrical field being continuous across the whole photoactive layer at short circuit. The small space-charge layer width is thus correlated with PCDTBT devices showing a more severe dependence of $J_{sc}$ upon photoactive layer thickness than the DT-PDPP2T-TT devices.

It follows from the discussion above that it appears likely that the loss of device performance with thick photoactive layers observed herein results primarily from the formation of a space-charge layer. Such space-charge layers can be caused by imbalanced and low charge carrier mobilities[49–51], as well as by asymmetric contact barriers or unintentional doping[15,19,20]. Regarding imbalanced mobilities, PCDTBT:PC$_{71}$BM blends, which show the strongest drop off of $J_{sc}$ with thickness, have been reported to show reasonably balanced electron and hole mobilities, indicating that imbalanced mobilities are unlikely to be the primary cause of the behaviour reported herein (see below for further discussion of the potential further impact of imbalanced mobilities)[52,53]. Note that the devices herein have identical contacts, with no evidence of significant surface recombination losses associated with contact barriers. As such, we focus herein on alternative origins of depletion layer formation.

Device band diagram simulations are shown in Fig. 4, including the potential impact of dark doping on band bending in the photoactive layer is shown in Fig. 4b (under illumination) and 4 f (in the dark). From these simulations, we find that a dark doping density of circa $3 \times 10^{16}$ cm$^{-3}$ is required to yield a depletion width of 100 nm. However work-function measurements (Fig. 5a, b) indicate dark doping densities of $10^{12}$–$10^{14}$ cm$^{-3}$ (calculation described in Supplementary Note 4) in the blends, independent of the substrate as shown in Fig. 5b, studied herein, orders of magnitude too low to yield the depletion layer widths measured in this study. This conclusion, also supported by SCLC analyses (see Supplementary Note 4 and Supplementary Fig. 12)[54], indicates unintentional photoactive layer dark doping cannot explain the thickness dependencies of $J_{sc}$ observed herein. We note that we also undertook Mott–Schottky analyses of the apparent doping densities of these devices calculated by admittance using the capacitance–voltage response at 1 kHz as plotted in

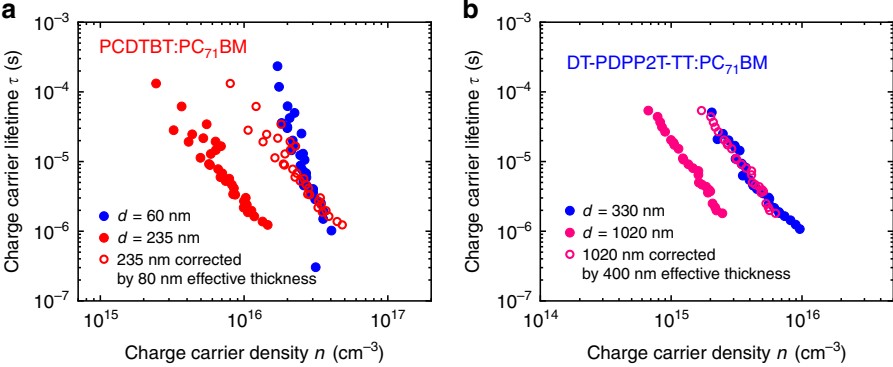

**Fig. 3** Evaluating the effective thickness. Recombination kinetics analysis of thick and thin devices of (**a**) PCDTBT:PC$_{71}$BM devices and (**b**) DT-PDPP2T-TT:PC$_{71}$BM, indicative of the effective thickness around 80 nm and 400 nm, respectively

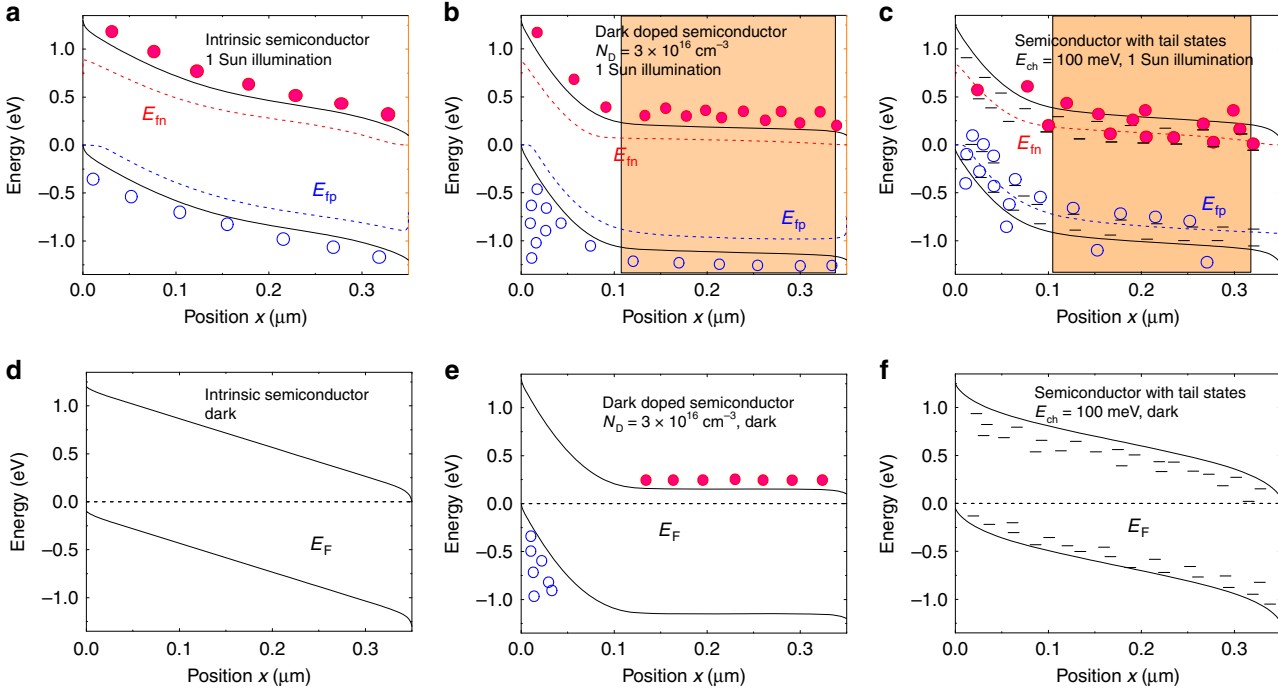

**Fig. 4** Band diagram simulations under one sun irradiation and in the dark condition. Band diagram simulations of the photoactive layer: (**a**) shows the intrinsic semiconductor under one sun illumination and (**d**) shows the same photoactive layer under dark conditions; (**b**) shows the photoactive layer with an electron doping level of $3 \times 10^{16}$ cm$^{-3}$ and (**e**) shows the same semiconductor under dark conditions; (**c**) demonstrates the photoactive layer with tail states with a total tail state density of $2 \times 10^{17}$ cm$^{-3}$ and a tail state slope $E_{ch}$ equals to 100 meV under one sun illumination condition, the illumination is from the right side (cathode here) correlating to the inverted device structure as used in PCDTBT and DT-PDPP2T-TT devices; and (**f**) shows the same junction but under dark condition. For (**b**) and (**c**), the orange region indicates the low field region in the junction where the collection efficiency is poor. Simulation parameters are shown is Supplementary Table 1. Error bars are the standard deviation of a minimum of four different measurements of different sample positions

Supplementary Fig. 13. These analyses gave much higher apparent doping densities than the dark work function and SCLC measurements discussed above, with PCDTBT devices showing apparently the highest doping density. However, SCAPS[55–57], simulations of these Mott–Schottky analyses in the presence of tail states indicate that such apparently high doping densities may result not only from ionisable dopants, but also from charge accumulation in tail states (see Supplementary Fig. 14 for details of these simulations).

**Impact of tail states on device performance**. As our analyses above have ruled out the presence of high ionised dopant densities as the origin of the limited depletion width we observe for

PCDTBT devices, we now consider the potential impact of low carrier mobilities on this width, focusing in particular upon the presence of tail states in limiting effective carrier mobilities. Device simulations in the presence of a tail of shallow traps are shown in the Fig. 4, in the dark (Fig. 4e) and under illumination (Fig. 4c), assuming for illustration an exponential distribution of tail states with total density $2 \times 10^{17}$ cm$^{-3}$ and characteristic energy $E_{ch} = 100$ meV, which is consistent with typical experimental assays of tail states in high disorder organic semiconductors (detailed simulation parameters are listed in Supplementary Table 1). It is apparent that under one sun irradiation, charge accumulation in these tail states results in significant band bending associated with the accumulation of space charge in these tail states, resulting in the generation of strong

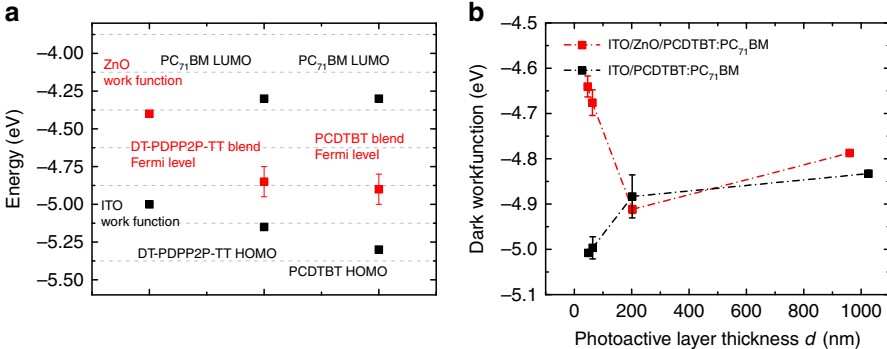

**Fig. 5** Energy level measurements. **a** Dark work-function and LUMO/HOMO values of 1 μm thick PCDTBT:$PC_{71}$BM and original DT-DPP2T-TT:$PC_{71}$BM films. **b** Thickness-dependent Fermi-level with different contacts of PCDTBT:$PC_{71}$BM films, a large change in the work function below 200 nm, but almost consistent over 200 nm

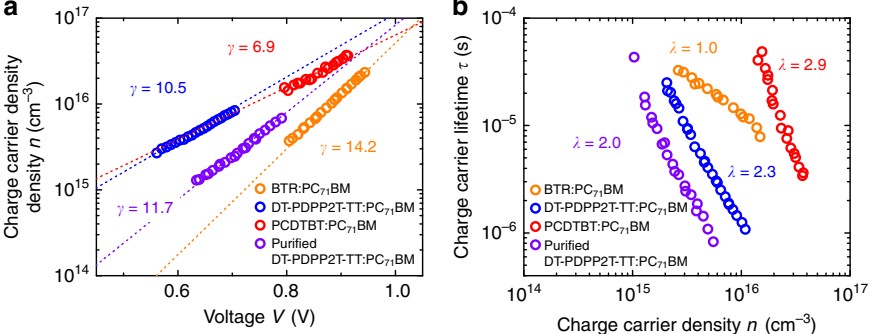

**Fig. 6** Effects of tail states on energetics and recombination kinetics. A comparison of energetics and recombination kinetics between BTR:$PC_{71}$BM, original DT-PDPP2T-TT:$PC_{71}$BM, PCDTBT:$PC_{71}$BM and purified DT-PDPP2T-TT:$PC_{71}$BM blends. (**a**) shows the charge carrier density as a function of photo-induced open-circuit voltage in the four material blend systems, the slope γ indicates the density of tail states distribution against the energy; (**b**) shows the charge carrier lifetime as a function of charge carrier density, the slope λ associates with the recombination reaction order δ in a correlation of $\delta = \lambda + 1$

electric field and low electric field regions analogous to those generated by dark doping. This simulation thus indicates the densities and energetics of tail states typically found in organic semiconductors are sufficient to result, under irradiation, in an effective space-charge layer width which could, for thick photo-active layers, limit photocurrent generation in organic solar cells (see below for discussion of further simulations as a function of illumination direction and electron/hole mobilities).

In order to investigate further, we studied the experimental correlation between the tail state density of states and the thickness dependence of device photocurrent collection. Determination of absolute densities of tail states in organic semiconductors is challenging, as this requires determination of the mobility edge which marks the transition from localised tail states to delocalised conduction/valence band states. However, CE measurements of charge density $n$ versus $V_{oc}$ provide a direct measurement of the energetic distribution of these tail states, quantified by their characteristic energy $E_{ch}$. The $E_{ch}$ derives from the slope γ of charge carrier density measured at different photo-induced $V_{oc}$ via $E_{ch} = 1/2\gamma$[58]. Typical data are shown in Fig. 6a, which yield values of $E_{ch}$ of 75 meV and 48 meV for PCDTBT and DT-PDPP2T-TT, respectively. Further data were also collected for BTR and purified DT-PDPP2T-TT, yielding values of $E_{ch}$ of 35 meV and 43 meV, respectively. Additional evidence for this trend of $E_{ch}$ comes from consideration of TPV measurements of carrier lifetime versus charge density, where the slope of these dependencies can be used to determine the characteristic energy of the tail state distribution determining this trapping/de-trapping mediated recombination behaviour

(indicated by the reaction order δ), as shown in Fig. 6b, and the effect of tail states on recombination has been previously discussed[32].

The trend of tail state density/energetics in this materials series was also investigated by ambient photoelectron spectroscopy (APS) measurements shown in Supplementary Fig. 16, where deviation from ideal behaviour at low energies is suggested to be indicative of tail state near the valence band-edge. These data indicate the relative distribution of tails states in the order BTR less than DT-PDPP2T-TT less than PCDTBT. Overall these data align very well with our $E_{ch}$ data. It is apparent that there is a clear correlation between values of $E_{ch}$ determined from these data and the thickness dependence of $J_{sc}$, with low energetic distributions of tail states correlating with an increase in the maximum thickness for efficient photocurrent. We can thus assign the variation in the maximum thickness for efficient photocurrent collection for the four device-series studied in Fig. 1 primarily to the impact of differing tail state distributions in the photoactive layers of these devices.

The presence of tail states is one factor which can limit the effective carrier mobilities of organic semiconductors. Assuming that charge carriers are immobile in the tail states ($n_t$) and mobile while above the mobility edge ($n_{free}$), transport can be described using multiple trapping and release model. In this model, charge carriers in the tail state can hop between different states via thermal activation, while charge carriers above the mobility edge can have a band-like transport with band mobility ($\mu_{free}$). The correlation between the effective mobility ($\mu^*$) and the trapped charge carriers ($n_t$) thus can be described as $\frac{n_{free}}{n_{free}+n_t} = \frac{\mu^*}{\mu_{free}}$[59]. Space-

charge layer limitations associated with photoinduced charge accumulation, as reported herein, can result from low effective carrier mobilities associated with either charge trapping in tail states, the primary focus of the study herein, or from low band mobilities. In this regard, it is interesting to note that the BTR devices show the narrowest distribution of tail states, but a $L_{max}$ less than for the purified DT-PDPP2T-TT devices. However, the BTR devices also showed lower effective mobility than the purified DT-PDPP2T-TT devices (Supplementary Fig. 15), which suggests that for the BTR devices, the lower $L_{max}$ most likely results from a lower band mobility rather than increased charge trapping in tail states, although more direct measurements of band mobility would be needed to confirm this point.

The challenge of achieving efficient device performance with thick photoactive layers appears particularly important for devices employing non-fullerene acceptors, where to date most studies have reported efficient performance only for absorber layer thicknesses less than 100 nm[60–63]. Our initial studies of this issue with IDTBR acceptor materials indicate high charge carrier densities stored in the device at open-circuit condition and significant wide distribution of tail states ($E_{ch}$ equals to 80 meV) as shown is Supplementary Fig. 17. This wide tail states distribution correlated with a significant loss of photocurrent density for IDTBR thicker devices (see Supplementary Fig. 18). Our preliminary studies for high mobility (See Supplementary Fig. 19) PBDB-T:ITIC and PBDTT-FTTE:m-ITIC devices are shown in Supplementary Fig. 17, indicating $E_{ch}$ equals to 63 and 77 meV, respectively, for the tail states distributions of devices. These values are indicative of a significant density of tail states likely to result in space-charge effects, consistent with reports that the optimal thickness for device performance is around 100 nm[62,63], without significant gain of photocurrent for thicker devices. As such, minimising tail state density for efficient, thick organic solar cells is likely to be a key issue for devices employing both fullerene acceptors, as studied herein, but also those employing non-fullerene acceptors.

The presence of tail states in organic semiconductors has to a range of origins, including variations in local film structure and material crystallinity as well as chemical defects and chain end groups[38,64–66]. In the study, we find that Soxhlet extraction of the DT-PDPP2T-TT polymer, which preferentially removes lower molecular weight polymer fractions both increase $L_{max}$ and reduces $E_{ch}$. This suggests that, for this polymer, tail states are associated with lower molecular weight polymer fractions, potentially associated with chain end groups. This conclusion is also consistent with the high $L_{max}$ obtained for the small molecular donor BTR which has a well defined structure, should have low/no (poly) dispersity and which does not include end groups[64,65]. This result also suggests that this purification strategy, and potentially other polymer synthesis/processing techniques which reduce the fraction of lower molecular weight polymer components, may be beneficial in enabling efficient photocurrent collection from thicker photoactive layers.

The impact of space-charge formation on the performance of thick OSC devices depends on whether this space-charge layer forms adjacent to the transparent contact or adjacent to the opposite, metal contact. In the latter case, photons will be absorbed primarily in the low field region, with consequently poor charge collection and a more severe thickness dependence. For matched carrier mobilities ($\mu_n = \mu_p$), and equal tail states distributions and densities for conduction and valence band tails, space charge will primarily accumulate adjacent to the metal contact, resulting in a field-free region adjacent to the transparent contact, and thus a significant loss of photocurrent. In these simulations, we use a simple optical model as described in Supplementary Table 1 based on Lambert–Beer's law which leads

to higher photogeneration closer to the illuminated surface of the solar cell. For example, in Fig. 4c, illumination is from the right-hand (cathode) side, and therefore holes have to travel a longer distance to reach the anode. The high photogeneration close to the cathode results in a high concentration of free and trapped electrons and holes essentially screening the electric field. While the electrons are readily extracted at the cathode, the holes diffuse to the anode where they will be present in excess relative to the electrons which neither diffuse to the anode (because of the repulsion by the built-in field) nor are they photogenerated close to the anode in large concentrations (because of Lambert–Beer's law and the assumed illumination direction). This large imbalance of hole to electron concentrations at the anode causes therefore positive space charge $\rho = q(p - n)$ and a substantial electric field. Note that based on this logic in the case of balanced mobilities and significant space charge, the field-free (low collection efficiency) region will unfortunately always coincide with the region of high photogeneration. As can be seen from the simulation in Supplementary Fig. 21(a), when the mobilities for electrons and holes are set to a relatively low value of $5 \times 10^{-5}$ cm$^2$ V$^{-1}$ s$^{-1}$ in the absence of tail states, no significant space-charge layer is apparent. In contrast, as shown in Supplementary Fig. 21 (b), if a tail state distribution with $E_{ch} = 100$ meV is added, a clear space-charge layer can be observed. In the case of mismatched mobilities, space-charge layer formation will tend to be dominated by the slower carrier, and its location will tend to be adjacent to the collecting contact for this slower carrier, (i.e. as shown in Supplementary Fig. 21(c) and (d), the space-charge layer locates at anode, where the slower carrier hole is collected.). This behaviour is likely to be the origin of the reasonably efficient device performance recently reported for thick polymer:IDTBR solar cells employing an inverted structure. In this case, the imbalanced mobilities ($\mu_h$ larger than $\mu_e$) present in this device will likely result in photoinduced electron accumulation at the cathode, thus localising the space-charge layer adjacent to the transparent contact, and therefore the primary region for light absorption[67]. In such a case, even though the photocurrent will not increase with thicker photoactive layers (due to the presence of an increasingly thick field-free region adjacent to the metal contact), device FF and $J_{sc}$ can be maintained, enabling device efficiency to be maintained for thick devices. We thus conclude that the presence of tail states and mismatched mobilities can both contribute to space-charge layer formation (see also Supplementary Fig. 21c and d), and can thus limit the effective photogenerated charge carrier collection width. We also note that the presence of tail states can themselves result in lower effective mobilities, further emphasising that these two considerations are interdependent. As such it can be concluded that consideration of both tail state densities and matched carrier densities will be important for the optimisation of thick organic solar cells.

## Discussion

We have investigated the dependence of photovoltaic performance on the absorber layer thickness for four different series of organic BHJ solar cells. The loss of performance for thick devices is found to be not only due to the loss of FF but also loss in $J_{sc}$. Previous studies on large absorber thickness devices have attributed the loss in performance, and particularly loss in FF, to a small mobility-lifetime product. While the loss of FF we observe is consistent with this conclusion, when we quantify the photocurrent loss resulting from the bimolecular (non-geminate) recombination flux at short circuit, we find this recombination loss to be too small to explain the loss of $J_{sc}$ for thick devices under one sun irradiation. Instead, we find that this photocurrent loss results from the formation of a space-charge layer which

results in a field-free region within the absorber layer where diffusion driven charge collection is inefficient. We observe that the formation of this space-charge layer requires the presence of tail (shallow trap) states in the absorber layer and depth-dependent photogeneration (stronger absorption close to the illuminated surface than deeper in the film). Comparing the four absorber layers studied, we find that a higher density/energetic distribution of tail states reduces the maximum thickness of the absorber layer which exhibits efficient photocurrent generation. Device simulation studies quantitatively support this conclusion, indicating that the presence of tail states can, combined with depth-dependent photogeneration, generate a space-charge layer which screens the built field in the device for thick absorber layers (See Supplementary Fig. 22). Purification of one polymer studied to remove lower molecular weight fractions is shown to both reduce the density of tail states, and increase the maximum thickness for efficient photocurrent generation. We therefore conclude that key considerations for the large scale printing of organic solar cells, which typically requires thick absorber layers (more than 300 nm), are not only a high mobility-lifetime product and matched mobilities but also that the density of tail states in the absorber layer should be minimised. The results and conclusions reported herein help explain the widely reported observation that many BHJs which perform well when spin coated as thin absorber layers, do not perform well when printed with thicker absorber layers.

## Methods

**Materials**. PCDTBT, DT-PDPP2T-TT and BTR were purchased from 1-Material. PC$_{71}$BM was purchased from Solenne BV. Chlorobenzene (CB), Chloroform (CF), O-dichlorobenzene (o-DCB), zinc acetate dihydrate, 2-methoxyethanol, and ethanolamine were purchased from Sigma Aldrich. All materials were used as received.

**Device fabrication**. ITO glass substrates were cleaned sequentially with deionised water, acetone, and isopropyl alcohol in an ultrasonic bath. PEDOT:PSS or ZnO precursor solution (109.8 mg of zinc acetate dehydrate dissolved in 1 ml of 2-methoxyethanol and 30.2 μl of ethanolamine) was spin-coated on plasma-treated ITO substrates at 4000 rpm for 40 s, followed by thermal annealing at 150 °C for 10 min. For electron only device, an electron injecting aluminium (AL) was thermal evaporated on ITO substrate with the thickness of 80 nm. For the hole only device, the hole injecting layer was used an 80 nm thermal evaporated gold (Au). PCDTBT and PC$_{71}$BM (1:2) were co-dissolved in CF at 50 °C and stirred at 600 rpm with a total concentration of 18 mg/ml, 30 mg/ml and 60 mg/ml for over 12 h. DT-PDPP2T-TT and PC$_{71}$BM (1:3) were co-dissolved in o-DCB with a total concentration of 7.5 mg/ml on an 80 °C hotplate stirred overnight in a nitrogen-filled glovebox. BTR and PC$_{71}$BM (1:1 weight ratio) were dissolved in CF with a total concentration of 40 mg/ml on a 60 °C hotplate stirred overnight in a nitrogen-filled glovebox. The blend solution was spin-coated on ZnO, PEDOT:PSS, Al or ITO glass inside the glovebox. The photoactive layer thicknesses were altering via varying spin rate or solution concentration. Finally, 10 nm of MoO$_3$ and 100 nm of silver for ZnO device (or 30 nm of calcium and 100 nm of aluminium) were thermally evaporated on the blend layer under the vacuum of $2 \times 10^{-5}$ mbar.

**J–V characterisations**. J–V response measurements were performed by a Keithley 2400 source meter under both solar simulators (Newport 92193A-1000) with the intensity of a 100 mW/cm$^2$ and fluorescent lamps (Osram L18W/827). The lux levels of the fluorescence lamps were measured by a lux meter, (LX-1330B).

**AFM measurements**. Atomic force microscopy (AFM) is a technique for studying surface topography of thin film samples. By using AFM three-dimensional information of the measured surface can be obtained with a resolution up to the scale of the atomic lattice. In the present study, all AFM measurements were carried out using an Agilent 5500 atomic force microscope in an ambient atmosphere. Atomic force microscopy study was carried out in tapping mode using an Agilent 5500 atomic force microscope in the ambient atmosphere. The approximate resonance frequency of the cantilever was 280 kHz and force constant was around 60 N m$^{-1}$.

**Photocurrent/CE/TPV measurements**. The corrected photocurrent $J_{ph}(V)$ was measured from the difference in J–V response between the dark response $J_{dark}(V)$ and the light response $J_{light}(V)$ by pulsed illumination to avoid overheating the samples. The pulsed illumination was provided by a ring of 12 white light-emitting

diodes (LEDs) with a fast-switching metal oxide semiconductor field-effect transistor. The one sun equivalent illumination was calibrated by matching the value of $J_{sc}$ and $V_{oc}$ obtained under the AM1.5G solar simulator measurement. The light was switched on for approximately 2 ms to allow a steady state to be reached and a much longer time with the light switched off was applied to avoid overheating. The potential bias was applied by a Keithley 2400 source meter, and the current and the voltage across were measured by a Tektronix TDS3032B Oscilloscope with a 1-MΩ input impedance. CE was used to determine the average charge carrier densities in devices under different illumination levels and different biases (open circuit and short circuit in this study). The desired light intensity was provided by a ring of 12 white LEDs same as described in corrected measurement capable of a power up to 5 sun equivalents. The device was held on under the initial bias at certain background light and then switch off to short circuit and turn off light, and the transient was acquired with a DAQ card connected to a Tektronix TDS3032B Oscilloscope. The voltage transients were converted into current transients through Ohms law, the current transients then were integrated to obtain total $Q$ to calculate $n$ in the device. During TPV measurements, the device was held at open-circuit condition under different background light intensity controlled by a ring of white LEDs as described before; then a small optical excitation was provided by a pulsed Continuum Minilite Nd:YAG laser at 532 nm with a pulse width of smaller than10nm. This small excitation produced a small voltage transient decay was then measured on the oscilloscope was fitted with a mono-exponential to obtain the small perturbation carrier lifetime and finally to be used to estimate the total charge carrier lifetime within the device.

**Kelvin probe and air photoemission spectroscopy**. Dark work function and air photoemission measurements were taken using an APS04 Air Photoemission system (APS04, by KP Technology) using a 2 mm gold tip under atmospheric conditions. All samples were measured on ITO substrates to ensure proper grounding of the organic thin films. Both APS and dark work-function measurements were taken at multiple positions on the films to ensure reproducibility. The work function of the tip was determined by comparison with a cleaned Ag reference. This was then used to convert contact potential differences between the tip and the sample to work function. The dark work function was determined after stabilisation of the signal, as initially, the contact potential difference was changing due to an ambient light-induced surface photovoltage. The APS data was processed using the protocol described by Baike et al.[68] This involves taking the cube root of the measured photoemission, fitting the resultant linear region and extrapolating to zero photoemission to find the HOMO level of the semiconductor.

**Device simulations** for Fig. 4 and Supplementary Fig. 20 and 21 employed the SCAPS software developed at TU Gent and which was previously reported in refs [55,69,70]. Simulation parameters are given in Supplementary Table 1.

## Data availability

The complete data set is available free of charge at http://zenodo.org (https://doi.org/10.5281/zenodo.2643248).

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

## Acknowledgements

The authors gratefully acknowledge the financial support of the UKRI Global Challenge Research Fund project SUNRISE (EP/P032591/1), the GIST Research Institute (GRI) grant funded by the GIST in 2019; the Global Research Laboratory Program of the National Research Foundation (NRF) funded by the Ministry of Science, ICT & Future Planning (NRF-2017K1A1A2013153); the GIST-ICL International Collaboration R&D Centre, the Welsh government funded Sêr Solar project, and the UK ESPRC for the Plastic Electronics Centre for Doctoral Training (EP/L016702/1) funding.

## Author contributions

TPV/CE, AFM, SCLC, J-V and Impedance measurements were carried out by J.W. Simulations were performed by J.W., T.K. Kelvin Probe and APS were carried out by J.L. Device fabrication were prepared by J.W., H.K. J.L., H.K., P.S.T. and H.C. Soxhlet extraction was carried out by S.-Y.J. W.C.T. and M.H. contributed to material preparation and sample fabrication. J.W., H.K., K. Lee, T.K., J.-S.K. and J.R.D. contributed to project planning and discussions. J.W., H.K. and J.R.D. prepared the manuscript. J.R.D. had the idea, led the project. All authors contributed to the manuscript preparation.

## Competing interests

The authors declare no competing interests.
