## [Peer Review File · Nature Communications]

Reviewers' comments:

Reviewer #1 (Remarks to the Author):

The manuscript by Wu et al addresses a very important topic in organic solar cells - the question of how to design photoactive systems that still retain good solar cell characteristics in the 200-300-nm thickness regime suitable for die-slot and related manufacturing. The main conclusions of this work is the loss in both FF and Jsc (which are really coupled rather than independent parameters) as film thickness increases is due not only to non-geminate recombination losses in the thicker films, but also to space-charge accumulation that cancels part of the internal field in the thicker films. The space-charge accumulation is suggested to occur preferentially at one contact.

This is phenomenologically the same as having an exiting layer of low carrier mobility. Such a situation has been discussed by Wagenpfahl et al in: Phys Rev B, 82 (2010)115306, and Tress et al in: AEM 3 (2013) 873. However no plausible explanation appears to have been offered for why such a layer would preferentially localize at one contact over a distance of 50 nm or so (Fig. 6). The tail states hypothesized in line 350-361 p 15, should equally well occur across the entire photoactive layer thickness.

However, more critically, the two issues of FF and Jsc losses that the authors studied are really not separate, but related by mobility. Insufficient mobility results in carrier pileup at one or both of the exiting contacts, resulting in a space-charge voltage penalty, which has been extensively discussed by Liu et al in: AEM (2013) doi 10.1002/aenm.201200972, and FF plots in: Nature Comms (2012) doi: 10.1038/ncomms2211. Incidentally, those authors also showed that the commonly assumed requirement for balanced carrier mobilities (mentioned by Wu et al in line 246-251, p 11) is predicated on the assumption of uniform photogeneration rates. For the thick devices that Wu et al study, this premise is false, and the requirement is for the carrier that travels the longer distance to be more (not as) mobile.

Therefore, while I like the topic that the authors are addressing very much, I regret I cannot recommend publication in present form. Would strongly encourage authors to revise their interpretation to take account of mobility on FF and Jsc. Other comments:

(i) Figure 2. The photogenerated current must equal collected current plus recombination current. Figure 2 attempts to show that this is not the case, which causes confusion. Clearly this means that the authors' accounting for recombination current is imperfect. Most likely explanation is the charge extraction/ transient photovoltage method used may be wrong.

(ii) Figure S7. Origin and assumptions in this plot do not appear to have been sufficiently described for readers to appreciate its validity. So the assertion that FF is separately well understood by the authors cannot yet be checked (lines 204-215, p 9). Note that such an assertion would also violate with the precept that FF and Jsc are in general coupled, as would be revealed by simple drift-diffusion-generation simulations.

(iii) Figure S9. Similarly, origin and assumptions in this plot do not appear to have been sufficiently described. So the assertion that there exists a significant depletion layer in the photoactive thickness cannot yet be checked. A leveling of capacitance was found for some photoactive layer systems and interpreted as evidence for a Schottky-depletion width of ca. 100 nm. Then the authors showed this cannot be the case, as amongst other reasons, KP measurements are not consistent with this doping level (Fig. 4) - FL is 0.25 - 0.5 eV away from HOMO band edge, so carrier density is roughly 10^{-4} to 10^{-8} of the effective DOS at band edge (not more than 10^{19} /cm³). Then the authors asserted

that the depletion width is in fact a trapped carrier density thickness due to tail states, which I couldn't understand. Following authors' arguments, the capacitance will now involve carriers are injected and stored inside this immobile layer. The device should not behave like a conventional dielectric (such as depletion dielectric), but a series capacitor with two different capacitances, so the interpretation of Fig S10 is complicated and not unique. In any case, what Fig 6 simulates is really the case of trapped carriers due to low carrier mobility in the vicinity of the hole contact. Simulation also does not give unique results, and on their own cannot provide proof of the carrier trapping localized to one contact of the photoactive layer... The most natural interpretation is to assume that the carriers are trapped across the photoactive layer, but as layer thickness increases, the pileup becomes increasing severe at the contact extracting the less mobile carrier, causing a severe space-charge voltage penalty there.

(iv) Figure 4. KP work function at 1 micron film thickness may not be reliable if the film behaves as insulation at this thickness.

(v) Figure 5. Discussion seems to be under construction?

Reviewer #2 (Remarks to the Author):

The loss of current density towards increased photoactive layer thickness is investigated and analyzed in this work. The authors conclude that it is the charge accumulation in intra-band tail state instead of the non-geminated recombination that should be responsible for the reduced current density in thick-film condition, because the smaller space-charge (depletion) layer width than active layer thickness is not beneficial for charge collection in the region outside, where the internal field is low. They also demonstrated that the depletion layer width is mainly determined by tail state properties, rather than by doping behavior. They further analyzed the origins of tail states, and suggested well defined structure and low polydispersity can help lowering the tail-state density and thus acquiring good thick-film performance. This manuscript is logically organized and the discussions are based on various material systems and valid characterizations. I think it may suitable for publication in the platform of Nature Communications if the following revisions can be fully addressed by the authors.

1) The conclusion could be debatable when considering the variation of film morphology, which is the issue that the authors overlooked throughout the whole manuscript. This issue should be carefully addressed, as the thick bulk-heterojunction film may present vertical distribution of the donor:acceptor that can significantly affect charge transport and extraction. If this is the case in the selected materials systems of this manuscript, the conclusion will be debatable.

2) The authors intended to extend the discussions to the non-fullerene systems, but failed to find a good combination that works well in thick-film condition. For instance, IDTBR system was selected, but the very low electron mobility of IDTBR makes it less representative in studying thick film devices. As a matter of fact, some non-fullerene systems based on ITIC derivatives reported recently indeed have good thick-film device performance (for example, Nat. Commun. 2018, 9, 743). In other words, as the photoactive materials used in the current manuscript are less representative than the widely used materials systems, such as the donor of P3HT, PCE10, and PBDB-T, and the molecular acceptor of ITIC, the generality and the influential of this work are weak. Thus, the authors should provide additional evidences of using these more widely used materials system to support their conclusions.

3) The authors mentioned that tail states are associated with lower molecular weight polymer fractions and chain end groups. However, both high-molecular-weight DT-PDPP2P-TT and small molecules BTR (low molecular weight) have low Ech. Even though dispersity is considered here, it remains unclear about the direct influence of molecular weight of polymers on the tail states. It will be helpful to systematically compare the "low molecular weight DT-PDPP2P-TT" (with molecular weight similar to

the fraction removed by Soxhlet extraction) to the original DT-PDPP2P-TT to clarify the effects of molecular weight. Moreover, the authors are also suggested to characterize the molecular weight and polydispersity of these samples by gel permeation chromatography.

4) When the authors introduces the promising long-term stable systems with efficiencies over 10% in page 3, some important work should be cited, e.g., Nat. Energy 2018, 3, 1051. When the authors introduces the materials systems work effectively with over 300 nm photoactive layer, some important work should be cited, e.g., Nat. Commun. 2014, 5, 5293.

5) Some mistakes in the "Device fabrication" section should be concerned, e.g., "30mf/ml", "DT-PDPP2P-TT and DT-PDPP2P-TT (1:3)", "MoO3". Please also care about the spacing between digital and unit, capital, and subscript throughout the main text and supporting information.

Reviewer #3 (Remarks to the Author):

The paper by Wu et al. is an interesting work and I enjoyed reading it. The authors combine many different electro-optical measurements with numerical simulations and the results are overall quite convincing. However, I have a few concerns that should be clarified by the authors.

There is abundant evidence that the low mobilities in organic semiconductors determine the FF and often the overall performance of organic solar cells. Several figures of merit have been proposed to describe the collection efficiency based on device and material parameters which were successfully applied to many OPV systems. One question in this regard is how the authors can disentangle the effects of these tail states from a low charge carrier mobility? In my understanding this shallow trapping will reduce the carrier mobility, i.e. the effects of tail states and mobility are perhaps quite similar. Is it possible to model how these tail states would impact a typical SCLC measurement and the effective mobility obtained from it? Related to that, the authors say that its unlikely that imbalanced mobilities are an issue in the PCDTBT blend, however, the $\mu\tau$ product of this blend is 2 orders of magnitude worse than in the DT-PDPP2P-TT blend which points to a mobility issue (there are also reports that show quite imbalanced mobilities in this system). Can it be said that $\mu\tau$ is not an issue (line 75, 76)? I assume the low field region in Figure 6c can be also reproduced with (10, 100x) imbalanced mobilities but no tail states? Perhaps it would be also nice to compare simulated JV curves with and without tail states to demonstrate their impact.

Figure S5 is nice which shows how much photocurrent can actually be extracted at far reverse biases, and also the comparison between J_{\max} and $J_{\text{SC}}+J_{\text{linearity,loss}}$. While I think the authors are probably correct that the linearity analysis of the short-circuit current cannot capture the whole non-geminate recombination loss (which indicates that there are additional first-order, non-geminate losses wrt the light intensity in these thick blends), the authors should be careful, whether they really capture the true linear region at low intensities, i.e. it would have been better to measure down to much lower intensities than 10% of a sun (e.g. ref. 34).

The authors should add how the characteristic energy E_{ch} is obtained from Figure 5a. I agree there is a correlation between E_{ch} and L_{\max} however purified DT-PDPP2P-TT has the largest L_{\max} but BTR the lowest E_{ch} . The change in E_{ch} upon purification is also rather small compared to the change in L_{\max} . The authors should comment on that.

Determination of depletion layer thickness is interesting and nicely done such as the evaluation of the photoactive layer doping density, energy levels and the simulations.

Minor points:

long-term stability could be quantified in the introduction.

Figure S3, P3HT seems most non-langevin in contrast to the caption.

Figure S8, how is the JV reconstructed?

Figure S9, some English issues in the caption.

Figure 4b, given that the workfunction of the thinner device (<100 nm) is closer to -5eV does that mean that there is more doping in the thinner layer ?

Table S1. It might be valuable to add how the release from tail states is treated.

Figure S14 problem on the x-axis

Figure S15, I am not sure if you can assign a slope to this "curves"

Response to the comments of reviewer

We would like to thank the referees for spending time on this manuscript and providing valuable comments which have greatly helped to improve the quality of the paper. The manuscript has been revised according to these comments as detailed below.

Reviewer 1

General comments: The manuscript by Wu et al addresses a very important topic in organic solar cells - the question of how to design photoactive systems that still retain good solar cell characteristics in the 200-300-nm thickness regime suitable for die-slot and related manufacturing. The main conclusions of this work is the loss in both FF and Jsc (which are really coupled rather than independent parameters) as film thickness increases is due not only to non-geminate recombination losses in the thicker films, but also to space-charge accumulation that cancels part of the internal field in the thicker films. The space-charge accumulation is suggested to occur preferentially at one contact. This is phenomenologically the same as having an exiting layer of low carrier mobility. Such a situation has been discussed by Wagenpfahl et al in: Phys Rev B, 82 (2010)115306, and Tress et al in: AEM 3 (2013) 873.

Our reply: Thanks for the reviewer's recommendation of two literatures.

We have included these reference papers on page 10 in the manuscript:

54. Tress, W. et al. Imbalanced mobilities causing S-shaped IV curves in planar heterojunction organic solar cells. *Appl. Phys. Lett.* 063301, 8–11 (2011).
55. Wagenpfahl, A. S-shaped current-voltage characteristics of organic solar devices. *Phys. Rev. B* 1–8 (2010). doi:10.1103/PhysRevB.82.115306
56. Tress, W., Corvers, S., Leo, K. & Riede, M. Investigation of Driving Forces for Charge Extraction in Organic Solar Cells : Transient Photocurrent Measurements on Solar Cells Showing S-Shaped Current – Voltage Characteristics. *Adv. Energy Mater.* 873–880 (2013). doi:10.1002/aenm.201200931'

However no plausible explanation appears to have been offered for why such a layer would preferentially localize at one contact over a distance of 50 nm or so (Fig. 6).

The tail states hypothesized in line 350-361 p 15, should equally well occur across the entire photoactive layer thickness.

Our reply: It is a very good point. We had briefly mentioned the location of the space charge layer appears in Figure 6 is associated with the non-uniform generation rate in the previous manuscript.

To further address this we have revised our manuscript on page 16-17 as :

‘For example, in Figure 6 (c), illumination is from the right hand (cathode) side, and therefore holes have to travel a longer distance to reach the anode, resulting in significant hole trapping and accumulation adjacent to the anode. Figure S20 (b) shows the reverse irradiation condition, where electrons have to travel a longer distance, resulting in electron accumulation on the cathode side.’

However, more critically, the two issues of FF and J_{sc} losses that the authors studied are really not separate, but related by mobility.

Our reply: As the reviewer correctly points out, both J_{sc} and FF are related to mobility. The way mobility affects J_{sc} and FF depends on the electrostatics: $\frac{\partial F}{\partial x} = \frac{q}{\epsilon} [p(x) - n(x)]$, the electric field distribution as a function of the spatial coordinate in the device. However, in our study we observed that FF and J_{sc} don't follow the same trend with photoactive layer thickness,. To further address the different dependence of J_{sc} (compared with FF) on device thickness, we have further included the simulation of photocurrent collected at short circuit condition which is based on the mobility-lifetime ($\mu\tau$) product (Hecht approximation) in the

supporting information. In the simulation, the two $\mu\tau$ product values ($3.4\text{E-}9 \text{ cm}^2/\text{Vs}$ and $1.9\text{E-}9 \text{ cm}^2/\text{Vs}$) are taken from the measurement of the DP-PDPP2P-TT and PCDTBT devices at short circuit condition. Here, we note that, in the simulation, the $\mu\tau$ product of $1.9\text{E-}9 \text{ cm}^2/\text{Vs}$ is sufficient enough to collect J_{max} with a 240 nm thick active layer. However, the PCDTBT device we measured, the J_{max} collected is within 100 nm thick active layer. Again, the mobility-lifetime product cannot explain the active layer thickness for J_{max} .

We have revised our manuscript on page 9-10 and added the simulation results in the SI as:

‘we take the mobilities and charge carrier lifetimes measured at short circuit condition, these yield the $\mu\tau$ products of $1.9\times 10^{-9} \text{ cm}^2 \text{ V}^{-1}$ and $3.4\times 10^{-9} \text{ cm}^2 \text{ V}^{-1}$ for PCDTBT and DT-PDPP2T-TT respectively. J_{sc} as a function of device thickness assuming a fully depleted device can then be simulated via the Hecht equation $J_{sc} = 2q\bar{G}\mu\tau \frac{V_{bi}}{d} \left(1 - \exp\left(-\frac{d^2}{2\mu\tau V_{bi}}\right)\right)$, as shown in Figure S9.^{21,50} These simulations indicate these $\mu\tau$ products yield maximal J_{sc} at 240 nm and 250 nm for the PCDTBT and DT-PDPP2T-TT devices respectively. This confirms our conclusion above that differences in charge carrier mobility and recombination assuming a fully depleted device can explain differences in the thickness dependence of FF between these devices, but cannot explain the difference in their thickness dependencies of J_{sc} .’

Figure S9. **Approximated J_{sc} based on the mobility-lifetime product as a function of active layer thickness.**

Insufficient mobility results in carrier pileup at one or both of the exiting contacts, resulting in a space-charge voltage penalty, which has been extensively discussed by Liu et al in: AEM (2013) doi 10.1002/aenm.201200972, and FF plots in: Nature Comms (2012) doi: 10.1038/ncomms2211.

Our reply: Thanks for the reviewer's suggestions on reference papers. We have added the references in Page 9:

52. Liu, B., Png, R. Q., Tan, J. K. & Ho, P. K. H. Evaluation of built-in potential and loss mechanisms at contacts in organic solar cells: Device model parameterization, validation, and prediction. Adv. Energy Mater. 4, (2014).

53. Liu, B. et al. High internal quantum efficiency in fullerene solar cells based on crosslinked polymer donor networks. Nat. Commun. 3, 1321–1328 (2012).

The point in the manuscript is that this pileup of charge can be caused by 1. Asymmetric and low mobilities; 2. Asymmetric contact barriers; 3. Unintentional dark doping, 4. Non-uniform generation rate and 5. The presence of tail states. Note that all of 1-4

can be amplified by 5 the presence of high densities of shallow defects to accumulate charge carriers to further aggravate the pileup as illustrated in Figure S21. The location of the space charge region depends on the effective slower carrier that reached to the collecting contact.

To further address this point, we have made the following changes (red) in our manuscript on page 11:

‘Such space charge layers can be caused by imbalanced and low charge carrier mobilities as well as by asymmetric contact barriers or unintentional doping... Note that the devices herein have identical contacts, with no evidence of significant surface recombination losses associated with contact barriers...’

Incidentally, those authors also showed that the commonly assumed requirement for balanced carrier mobilities (mentioned by Wu et al in line 246-251, p 11) is predicated on the assumption of uniform photogeneration rates. For the thick devices that Wu et al study, this premise is false, and the requirement is for the carrier that travels a longer distance to be more (not as) mobile.

Therefore, while I like the topic that the authors are addressing very much, I regret I cannot recommend publication in present form. Would strongly encourage authors to revise their interpretation to take account of mobility on FF and Jsc.

Our reply: A key point in our study is to address the impact of non-uniform generation as the uniform generation is the very ideal case. We have emphasised this further in our revised manuscript, adding the following text on pages 17 and additional figures in the SI:

‘As can be seen from the simulation in Figure S21 (a), when the mobilities for electrons and holes are set to a relatively low value of $5 \times 10^{-5} \text{ cm}^2 \text{V}^{-1} \text{s}^{-1}$ in the absence of tail states, no significant space charge layer is apparent. In contrast, as shown in Figure S21 (b), if a tail

state distribution with $E_{ch} = 100$ meV is added, a clear space charge layer can be observed. In the case of mismatched mobilities, space charge layer formation will tend to be dominated by the slower carrier, and its location will tend to be adjacent to the collecting contact for this slower carrier (i.e. as shown in Figure S21 (c) and (d), the space charge layer locates at anode, where the slower carrier-hole is collected.).’

As the reviewer has commented, the FF and J_{sc} are coupled and depend on mobility. However, in this work, we emphasise that FF and J_{sc} show different dependence of the devices we studied. In fact, the photocurrent generation should increase with absorber thickness (i.e. in the paper by Andrew etc. (DOI: 10. 1021/nm2002695),¹ in Figure 5 of this paper, the FF drops with active layer thickness, but the J_{sc} increase with active layer thickness), and we have confirmed this for the systems we studied using large reverse bias extraction. As described above, the main point in this paper is that the J_{sc} loss is not only associated with the loss of device FF in thick devices. To further emphasise this we have revised our paper on page 7:

‘Our observation of J_{sc} increasing with active layer thickness but FF decreasing has also been reported in the literature, and clearly indicates different dependencies of J_{sc} and FF upon active layer thickness.’

Comments 1: Figure 2. The photogenerated current must equal collected current plus recombination current. Figure 2 attempts to show that this is not the case, which causes confusion. Clearly this means that the authors' accounting for recombination current is imperfect. Most likely explanation is the charge extraction/ transient photovoltage method used may be wrong.

Our reply: As discussed in the manuscript, we use both TPV/CE and J_{sc} linearity analysis;

these methods use very different data and are based on different assumptions, but both yield the same result – that they are unable to explain the loss of J_{sc} with thickness. Our point is not that these analyses are fundamentally wrong, as they work well for thin devices, but that one or more of the assumptions underlying these models break down for thick devices. In particular, the TPV/CE analysis assumes a continuous electrical field across the whole photoactive layer at short circuit. The problem is that if we use this model to analysis some thick devices, as we illustrated in Figure 2, we will obtain a wrong recombination flux value. As we have published previously (ref 51, DOI: 10.1103/PhysRevX.5.011032), this analysis can be corrected if we use an effective thickness for the active light absorber layer. In order to emphasise this point more clearly, we have added the text in red on page 9:

‘We also note that these analyses were able to correctly reproduce the J-V curves of thin PCDTBT and DT-PDPP2T-TT devices assuming the photo-generation current density equals to J_{sc} (See Figure S8), **confirming the validity of these analyses for thin devices...**’

Comments 2: Figure S7. Origin and assumptions in this plot do not appear to have been sufficiently described for readers to appreciate its validity. So the assertion that FF is separately well understood by the authors cannot yet be checked (lines 204-215, p 9). Note that such an assertion would also violate with the precept that FF and J_{sc} are in general coupled, as would be revealed by simple drift-diffusion-generation simulations.

Our reply: FF is indeed a more complicated parameter than J_{sc} ; there are several papers which have demonstrated the link between the FF and the competition between the collection and recombination. There are several figures of merit quantifying FF such as mobility-lifetime product and non-Langevin factors, with both these figure of merits

agreeing with the FF trend observed in our study. Regarding J_{sc} , there is one approximation based on the mobility-lifetime product called the Hecht equation, we have now further included this analysis on page 9 and the Figure S9 (see above for details of this revision). Here we emphasise that we are able to quantify photocurrent loss at different operation bias using a fully depletion model with TPV/CE result for the relatively thin device as shown in Figure S8. This analysis breaks when the depletion width is smaller than the photoactive layer width.

Comments 3: Figure S9. Similarly, origin and assumptions in this plot do not appear to have been sufficiently described. So the assertion that there exists a significant depletion layer in the photoactive thickness cannot yet be checked. A leveling of capacitance was found for some photoactive layer systems and interpreted as evidence for a Schottky-depletion width of ca. 100 nm. Then the authors showed this cannot be the case, as amongst other reasons, KP measurements are not consistent with this doping level (Fig. 4) - FL is 0.25 - 0.5 eV away from HOMO band edge, so carrier density is roughly 10^{-4} to 10^{-8} of the effective DOS at band edge (not more than 10^{19} /cm³). Then the authors asserted that the depletion width is in fact a trapped carrier density thickness due to tail states, which I could not understand.

Our reply: As replied before, the formation of the space charge layer can be caused by 1. Asymmetric and low mobilities; 2. Asymmetric contact barriers; 3. Unintentional dark doping, 4. Non-uniform generation rate and 5. The presence of tail states. In this study, the example blends we used do not show 1. significant mobility mismatch and 2. contact barriers. Hence, we checked the unintentional dark doping level, which is validly determined by dark KP measurement. Regarding the Mott-Schottky analysis, which is based on voltage-

dependent capacitance measurement, even it is measured under dark, there is still a significant chemical capacitance caused by injection. In this case, it is not a straight forward way to determine the dark doping level in the blends.

To make a clearer discussion, we have revised (red part) our manuscript on page 11 as:

‘It follows from the discussion above that it appears likely that the loss of device performance with thick photoactive layers observed herein results primarily from the formation of a space charge layer. Such space charge layers can be caused by imbalanced and low charge carrier mobilities^{55–57} as well as by asymmetric contact barriers or unintentional doping.^{19,51,52}

Regarding imbalanced mobilities, PCDTBT:PC₇₁BM blends, which show the strongest drop off of J_{sc} with thickness, have been reported to show reasonably balanced electron and hole mobilities, indicating that imbalanced mobilities are unlikely to be the primary cause of the behaviour reported herein (see below for further discussion of the potential further impact of imbalanced mobilities).^{58,59} Note that the devices herein have identical contacts, with no evidence of significant surface recombination losses associated with contact barriers. As such, we focus herein on alternative origins of depletion layer formation, starting with consideration of unintentional photoactive layer doping.’

Following authors' arguments, the capacitance will now involve carriers are injected and stored inside this immobile layer. The device should not behave like a conventional dielectric (such as depletion dielectric), but a series capacitor with two different capacitances, so the interpretation of Fig S10 is complicated and not unique. In any case, what Fig 6 simulates is really the case of trapped carriers due to low carrier mobility in the vicinity of the hole contact. Simulation also does not give unique results, and on their own cannot provide proof of the carrier trapping localized to one contact of the photoactive layer... The most natural interpretation is to assume that the carriers are trapped across the

photoactive layer, but as layer thickness increases, the pileup becomes increasingly severe at the contact extracting the less mobile carrier, causing a severe space-charge voltage penalty there.

Our reply: In the simulations in our submitted manuscript, we assumed electrons and holes have the same mobility. As the generation rate near the illumination side is higher than the opposite contact, the average travel distance of the charge carrier that needs to be collected at the opposite contact is longer. Therefore, the space charge layer appears at the opposite side of the illumination (see Figure S20). We have now added simulations for the case of mobility mismatch, as shown in **Figure S21**, where the space charge layer appears at the collection contact for the slower charge carrier. We have now further emphasized that for both non-uniform generation, mobility mismatch and tail states can all contribute to the formation of space charge layers and field-free regions, as detailed above in our revisions on page 16 & 17.

Comments 4: Figure 4. KP work function at 1 micron film thickness may not be reliable if the film behaves as insulation at this thickness.

Our reply: The ~1 micron device is still working but with low device performance as shown below.

Comments 5: Discussion seems to be under construction?

Our reply: The discussion is on pages 15-17 of the manuscript.

Reviewer 2:

General comments: The loss of current density towards increased photoactive layer thickness is investigated and analyzed in this work. The authors conclude that it is the charge accumulation in intra-band tail state instead of the non-geminated recombination that should be responsible for the reduced current density in thick-film condition, because the smaller space-charge (depletion) layer width than active layer thickness is not beneficial for charge collection in the region outside, where the internal field is low. They also demonstrated that the depletion layer width is mainly determined by tail state properties, rather than by doping behavior. They further analyzed the origins of tail states, and suggested well defined structure and low polydispersity can help lowering the tail-state density and thus acquiring good thick-film performance. This manuscript is logically organized and the discussions are based on various material systems and valid

characterizations. I think it may be suitable for publication in the platform of Nature Communications if the following revisions can be fully addressed by the authors.

Comments 1: The conclusion could be debatable when considering the variation of film morphology, which is the issue that the authors overlooked throughout the whole manuscript. This issue should be carefully addressed, as the thick bulk-heterojunction film may present vertical distribution of the donor:acceptor that can significantly affect charge transport and extraction. If this is the case in the selected materials systems of this manuscript, the conclusion will be debatable.

Our reply: The reviewer is correct that vertical composition variations can impact upon device performance, although conclusions on their importance differ significantly in the literature. Such gradients in composition (or morphology) are most likely to impact upon the thickness dependence of FF rather than J_{sc} and therefore unlikely to be important for the results and conclusions reported in this manuscript. As such, we have added the following sentences to the manuscript on page 7 to discuss this point.

‘Our observation of J_{sc} increasing with active layer thickness but FF decreasing has also been reported in the literature, and clearly indicates different dependencies of J_{sc} and FF upon active layer thickness.’

Comments 2: The authors intended to extend the discussions to the non-fullerene systems, but failed to find a good combination that works well in thick-film condition. For instance, IDTBR system was selected, but the very low electron mobility of IDTBR makes it less

representative in studying thick film devices. As a matter of fact, some non-fullerene systems based on ITIC derivatives reported recently indeed have good thick-film device performance (for example, Nat. Commun. 2018, 9, 743). In other words, as the photoactive materials used in the current manuscript are less representative than the widely used materials systems, such as the donor of P3HT, PCE10, and PBDB-T, and the molecular acceptor of ITIC, the generality and the influential of this work are weak. Thus, the authors should provide additional evidences of using these more widely used materials system to support their conclusions.

Our reply: The reviewer is correct that our discussion of blends with NFA's was very limited in our submitted manuscript. We have now added further data on the tail state distribution and the effective mobility of blends with the NFA's ITIC and m-ITIC in the supporting information. Our preliminary studies for PBDB-T:ITIC and PBDTT-FTTE:m-ITIC devices are shown to have $E_{ch} \sim 63$ and 77 meV of tail states distribution as shown in Figure S17. The effective mobility of PBDB-T:ITIC is higher than the PBDTT-FTTE:O-IDTBR as shown in Figure S18. As reported in the thickness dependent studies, the optimal thickness is still limited by ~ 100 nm (DOI: 10.1002/adma.201703527; 10.1021/jacs.6b09110),^{2,3} without significant gain of photocurrent at 200 – 300 nm device thickness.

We have revised our manuscript on Page 15:

‘Our preliminary studies for high mobility (See Figure S19) PBDB-T:ITIC and PBDTT-FTTE:m-ITIC devices are shown in Figure S17, indicating $E_{ch} \sim 63$ and 77 meV respectively for the tail states distributions of devices. These values are indicative of significant densities of tail states likely to result in space charge effects, consistent with reports that the optimal thickness for device performance is ~ 100 nm,^{65,66} without significant gain of photocurrent for thicker devices.’

The NFAs based devices have added in SI:

Figure S17. Energetics of non-fullerene based PBDTT-FTTE:O-IDTBR, PBDTT-FTTE:EH-IDTBR, PBDB-T:ITIC, PBDTT-FTTE:m-ITIC and PffBT4T-2OD:EH-IDTBR devices.

The charge carrier density as a function of voltage plots shows a slope at the range of 6.3 to 7.9 in these non-fullerene acceptor based device systems (the device thicknesses range between 70 to 115 nm). The slope γ is from the exponential fit $n = n_0 \exp(\gamma V_{oc})$ of the measured charge carrier density as a function of V_{oc} . The tail state slope E_{ch} derives via $E_{ch} = 1/(2\gamma)$ using the charge extraction results measured at open circuit.⁴

Figure S19. Effective mobility of non-fullerene based PBDTT-FTTE:O-IDTBR, PBDB-T:ITIC and PBDTT-FTTE:m-ITIC devices.

Comments 3: The authors mentioned that tail states are associated with lower molecular weight polymer fractions and chain end groups. However, both high-molecular-weight DT-PDPP2T-TT and small molecules BTR (low molecular weight) have low Ech. Even though dispersity is considered here, it remains unclear about the direct influence of molecular weight of polymers on the tail states. It will be helpful to systematically compare the “low molecular weight DT-PDPP2T-TT” (with molecular weight similar to the fraction removed by Soxhlet extraction) to the original DT-PDPP2T-TT to clarify the effects of molecular weight. Moreover, the authors are also suggested to characterize the molecular weight and polydispersity of these samples by gel permeation chromatography.

Our reply: The end group of polymer might introduce more energetic disorder, as we cited already in the manuscript, the selection of high molecular weight part of the polymer is

beneficial for device performance.(DOI: 10.1038/ncomms6688)⁵⁻⁷ Hence, we proposed the large molecular weight, and it proved to be right in the DT-PDPP2T-TT case. The energetic disorder of small molecule materials is more sensitive to its morphology. Here the efficient thick BTR device requires an additional solvent vapour anneal process to reach the morphology with a low energetic disorder (10.1039/c7ta10875c).⁸ The energetic disorder itself is a complicated topic in organic materials it is strongly associated with the chemical structure and the packing of the material as well as the blend morphology, etc. Therefore, we proposed two possible way to reduce energetic disorder.

Comments 4: When the authors introduces the promising long-term stable systems with efficiencies over 10% in page 3, some important work should be cited, e.g., Nat. Energy 2018, 3, 1051. When the authors introduces the materials systems work effectively with over 300 nm photoactive layer, some important work should be cited, e.g., Nat. Commun. 2014, 5, 5293.

Our reply: These papers suggested by the review are very relevant. We have added these important work as our references.

6. Fan, B. et al. Fine-tuning of the chemical structure of photoactive materials for highly efficient organic photovoltaics. *Nat. Energy* 3, (2018).
13. Liu, Y. *et al.* Aggregation and morphology control enables multiple cases of high-efficiency polymer solar cells. *Nat. Commun.* **5**, 1–8 (2014).

Comments 5: Some mistakes in the “Device fabrication” section should be concerned, e.g., “30mf/ml”, “DT-PDPP2T-TT and DT-PDPP2T-TT (1:3)”, “MoO3”. Please also care about the spacing between digital and unit, capital, and subscript throughout the main text and supporting information.

Our reply: Thanks for the reviewer's corrections. We have corrected these mistakes in our manuscript on page 19.

Reviewer#3(Remarks to the Author):

The paper by Wu et al. is an interesting work and I enjoyed reading it. The authors combine many different electro-optical measurements with numerical simulations and the results are overall quite convincing. However, I have a few concerns that should be clarified by the authors.

There is abundant evidence that the low mobilities in organic semiconductors determine the FF and often the overall performance of organic solar cells. Several figures of merit have been proposed to describe the collection efficiency based on device and material parameters which were successfully applied to many OPV systems. One question in this regard is how the authors can disentangle the effects of these tail states from a low charge carrier mobility?

Our reply: Thanks for the reviewer raising this very valuable question. Theoretically, the effective mobility can be caused by low band mobility, and such low band mobility can also contribute to the formation of the space charge layer. To experimentally disentangle the effect of low band mobility on the low effective mobility, temperature dependent measurements are necessary but beyond the scope of our present study. Moreover, the effect of tail states on the low effective mobility can be assessed via theory, thus we have revised our manuscript and

further addressed the effect of tail states on charge carrier accumulation based on the effects of tail states on the low effective mobilities on page 15 as below:

The low mobilities of organic materials are also associated with the presence of tail states.

Assuming that charge carriers are immobile in the tail states (n_t) and mobile while above the mobility edge (n_{free}), the transport can be described using a multiple trapping and release model. In this model, charge carriers in the tail state can hop between different states via thermal activation, while charge carriers above the mobility edge can have a band-like transport with a band mobility (μ_{free}). The correlation between the effective mobility (μ^*) and

the trapped charge carriers (n_t) thus can be described as $\frac{n_{free}}{n_{free}+n_t} = \frac{\mu^*}{\mu_{free}}$.⁶⁴ Assuming the

short circuit current density to be drift-dominated as $J_{sc} = \mu_{drift} n F$, where the n is the charge carrier density and F is the electric field, for the system without tail states in which all the charge carriers are delocalised, the current density depends only on the band mobility (μ_{free}) and the density of free charge carriers (n_{free}) via $J_{sc} = \mu_{free} n_{free} F$. Contrary to the tail states free system, a system with tail states, in which the effective mobility relates to both free and trapped charge carriers $n_{free} + n_t = n_{total}$, and where most charge carriers are localised $n_t \gg n_{free}$, the short circuit current density can thus be described as $J_{sc} = \mu^* n_{total} F$. In this circumstance, the effective mobility of the systems with tail states is much lower (assuming identical J_{sc} and F) because most charge carriers are trapped and immobile ($n_t \gg n_{free}$).

Thus, for lower effective mobilities caused by charge storage in band tails, the concentration of trapped and immobile charge at short circuit can be vastly increased. This charge can then in turn affect the electric field distribution within the active layer.

In my understanding this shallow trapping will reduce the carrier mobility, i.e. the effects of tail states and mobility are perhaps quite similar. Is it possible to model how these tail states

would impact a typical SCLC measurement and the effective mobility obtained from it?

Our reply: It is true as the reviewer mentioned, the shallow traps will slow down charge carrier transport, for example, our previous study has reported that the spread of tail state distribution reduces the effective mobility of BTR:PC₇₁BM device.⁸ However, the effective mobility is also affected by the ‘intrinsic material mobility’, e.g. the DT-PDPP2T-TT shows a wider distribution of tail states than BTR, but its effective mobility is higher than BTR (see new added Figure S15). For effective mobility measured by this charge extraction method, we cannot tell the difference between electron and hole, as charge extraction is performed on the full device. As the reviewer suggested, it will be very important to separate the effect of hole and electron transport. We tried SCLC analysis, however, we were stuck by the analysis of the SCLC fitting as we sometimes found the fitting can result in a several orders of difference in the analysed mobilities.

We have further included the effective mobility of the devices shown in the manuscript:

Figure S15. Effective charge carrier mobility for PCDTBT:PC₇₁BM (red) BTR:PC₇₁BM (green), DT-PDPP2T-TT:PC₇₁BM (blue) and purified DT-PDPP2T-TT:PC₇₁BM (violet),

measured by charge extraction at short circuit condition.

Related to that, the authors say that its unlikely that imbalanced mobilities are an issue in the PCDTBT blend, however, the $\mu\tau$ product of this blend is 2 orders of magnitude worse than in the DT-PDPP2T-TT blend which points to a mobility issue (there are also reports that show quite imbalanced mobilities in this system). Can it be said that $\mu\tau$ is not an issue (line 75, 76)?

Our reply: The referee is quite correct that the $\mu\tau$ product is two orders of magnitude less for the PCDTBT device and the DT_PDPP2P-TT device, as we determine directly for these devices at MPP in Figure S7. This agrees with the differing dependencies of device FF on thickness. However, when determined at short circuit (conditions relevant to J_{sc} , the $\mu\tau$ difference becomes much less (3.4E-9 cm²/Vs for PDPP2P-TT and 1.9E-9 cm²/Vs for PCDTBT). When we undertake a J_{sc} simulation based on $\mu\tau$ using the Hecht equation, as now added in Figure S9, this difference in the product is far too small to explain the differing dependencies of J_{sc} upon thickness. We have now revised our manuscript to address this point on page 9-10 in the manuscript as follows:

' we take the mobilities and charge carrier lifetimes measured at short circuit condition, these yield the $\mu\tau$ products of $1.9 \times 10^{-9} \text{ cm}^2 \text{ V}^{-1}$ and $3.4 \times 10^{-9} \text{ cm}^2 \text{ V}^{-1}$ for PCDTBT and DT-PDPP2T-TT respectively. J_{sc} as a function of device thickness assuming a fully depleted device can then be simulated via the Hecht equation $J_{sc} = 2q\bar{G}\mu\tau \frac{V_{bi}}{d} \left(1 - \exp\left(-\frac{d^2}{2\mu\tau V_{bi}}\right)\right)$, as shown in Figure S9.^{21,50} These simulations indicate these $\mu\tau$ products yield maximal J_{sc} at 240 nm and 250 nm for the PCDTBT and DT-PDPP2T-TT devices respectively. This confirms our conclusion above that differences in charge carrier mobility and recombination

assuming a fully depleted device can explain differences in the thickness dependence of FF between these devices, but cannot explain the difference in their thickness dependencies of J_{sc} .

And have included the simulation in supporting information.

Figure S9. Approximated J_{sc} based on the mobility-lifetime product as a function of active layer thickness.

I assume the low field region in Figure 6c can be also reproduced with (10, 100x) imbalanced mobilities but no tail states?

Our reply: Yes, it is true. As shown in Figure S21 (c) (we have included additional 4 SI figures the order of this figure have been changed), the space charge layer formed if the device shows imbalanced mobility. Figure S21 (d) shows the band diagram of the device with imbalanced mobility (same as Figure S21(c) and with additional tail states. The presence of additional tail states will further aggravate the charge carrier accumulation if there is imbalanced mobility.

Figure S21. Band diagrams of 350nm thick solar cells. Device with different tail states distribution and electron/hole mobilities are shown separately in (a) $E_{ch} = 30 \text{ meV}$, $\mu_e = \mu_h$ (b), $E_{ch} = 100 \text{ meV}$, $\mu_e = \mu_h$, (c) $E_{ch} = 30 \text{ meV}$, $\mu_e = 10\mu_h$, (d) $E_{ch} = 100 \text{ meV}$, $\mu_e = 10\mu_h$. Fitting parameters used are listed in Table S1.

Perhaps it would be also nice to compare simulated JV curves with and without tail states to demonstrate their impact.

Our reply: It is very hard to do this. The effects of tail states on the device performance are complex, which can affect properties such as effective mobility, recombination and charge carrier distribution at the same time. As our simulation is based on the experimental data rather than an independent model, we are afraid of having to change too many parameters

at the same time in order to simulate the effect of tail states on JV response.

Figure S5 is nice which shows how much photocurrent can actually be extracted at far reverse biases, and also the comparison between J_{max} and $J_{\text{SC}}+J_{\text{linearity,loss}}$. While I think the authors are probably correct that the linearity analysis of the short-circuit current cannot capture the whole non-geminate recombination loss (which indicates that there are additional first-order, non-geminate losses wrt the light intensity in these thick blends), the authors should be careful, whether they really capture the true linear region at low intensities, i.e. it would have been better to measure down to much lower intensities than 10% of a sun (e.g. ref. 34).

Our reply: As the reviewer suggested it will nice to look the J_{sc} down to lower light levels. Actually we did measurement start from 0.1% Sun, at a linear-linear scale, there is no visible difference between 0.1% to 10% Sun. However, when we calculate the derivation of $d\ln J_{\text{sc}}/d\ln \Phi$, there is large noise between 0.1% to 10% Sun due to the error of light intensity which is limited by our light source. So, in the derivation analysis, we normally ignore the data points that below to 10% with our present set-up.

The authors should add how the characteristic energy E_{ch} is obtained from Figure 5a. I agree there is a correlation between E_{ch} and L_{max} however purified DT-PDPP2T-TT has the largest L_{max} but BTR the lowest E_{ch} . The change in E_{ch} upon purification is also rather small compared to the change in L_{max} . The authors should comment on that.

Our reply: Thanks for the reviewer's comments. It has been discussed in Kirchartz' paper (DOI: 10.1103/PhysRevB.86.165201) on the relationship between γ (charge extraction slope at open circuit) and E_{ch} in Eq. (9),⁴ and we have cited this in the manuscript. Regarding the difference of L_{max} between DT-PDPP2T-TT and BTR, it is probably something to do with mobility. As the DT-PDPP2T-TT shows higher mobility than BTR device, therefore the de-trapping process is faster. We have revised this point in the manuscript and include the mobility data in Figure S15.

We have added how to assess the E_{ch} from CE measurement on page 7:

'The E_{ch} derives from the slope γ of charge carrier density measured at different photo-induced Voc via $E_{ch}=1/2\gamma$.⁶³'

We have made the corresponding conclusion in the manuscript on page 14-15 as:

'Here we note that the BTR device shows the most narrow distribution of tail states, but the L_{max} is less than DT-PDPP2T-TT device, which may be related to the lower mobility of the charge carriers in the BTR device (Figure S15), which is likely to further increase charge accumulation in this device.'

The effective mobility data have also added in the supporting information:

Figure S15. Effective charge carrier mobility for PCDTBT:PC₇₁BM (red) BTR:PC₇₁BM (green), DT-PDPP2T-TT:PC₇₁BM (blue) and purified DT-PDPP2T-TT:PC₇₁BM (violet).

Determination of depletion layer thickness is interesting and nicely done such as the evaluation of the photoactive layer doping density, energy levels and the simulations.

Minor points:

long-term stability could be quantified in the introduction.

Figure S3, P3HT seems most non-langevin in contrast to the caption.

Reply: Thanks for the reviewer's comments. We have corrected the mistake in the supporting information.

Figure S3. **Non-Langevin factors of different organic bulk heterojunction devices.** Except for P3HT:PC₇₁BM device, the other bulk heterojunction solar cells show relatively Langevin recombination. Non-langevin factors determined from the quotient of the effective bimolecular recombination coefficient k_{bi} (measured by TPV/CE) over the Langevin recombination coefficient $k_{langevin}$.

Figure S8, how is the JV reconstructed?

Reply: We use the model of in the steady state, the current collect equals to the current generated substrates the current loss through non-geminate recombination: $J = J_{gen} - J_{loss}$.⁹ The non-geminate recombination is measured using TPV/CE. We have added further note on to do J-V reconstruction using TPV/CE results in SI as:

The current density can be described by the continuity equation at every operating point across the voltage range:

$$\frac{1}{q} \frac{dJ}{dx} + G - R = 0 \quad 1$$

J is current density, q is the elementary charge, x is the spatial location of the device, G is generation rate in volume, R is recombination rate.

Using the charge extraction across the J-V, the charge carrier density at different operating bias can be assessed. Therefore, the integration of Eq.1 across the active layer thickness we

can obtain the J-V response:

$$J(V) = -qG(V)d + qdR(V) \quad 2$$

If the generation is field independent, $G(V)$ is independent to bias therefore is a constant, here we use $qG(V)d = J_{ph}^{saturated}$. $qdR(V)$ is the recombination flux at different bias conditions, where the recombination rate $R(V)$ is defined as $R(V)=k_{bi}(n(V))^2$, k_{bi} is effective bimolecular recombination coefficient calculated by TPV/CE results measured at open circuit.

Figure S9, some English issues in the caption.

Reply: Thank for the reviewer's comments. We have corrected the caption.

Figure 4b, given that the workfunction of the thinner device (<100 nm) is closer to -5eV does that mean that there is more doping in the thinner layer?

Reply: Not necessarily. The band bending changes with different bottom substrate which indicates injection from the contact rather than dark doping.

Table S1. It might be valuable to add how the release from tail states is treated.

A single tail state can capture and emit a hole (or electron), in thermal equilibrium the occupation function for all charge carriers (free or trapped, electrons or holes) must be the Fermi-Dirac function in thermal equilibrium. The occupation of traps in the simulation is widely used with the Shockley-Read-Hall statistics (DOI: 10.1103/PhysRev.87.387; 10.1103/PhysRev.87.835).^{10,11}

Figure S14 problem on the x-axis

Reply: Thanks for the reviewer's correction. We have replotted the graph.

Figure S16. **Ambient Photoemission Spectroscopy results.** The PCDTBT:PC₇₁BM film (red) shows the largest tail area comparing with BTR:PC₇₁BM (green) and DT-PDPP2T-TT:PC₇₁BM (blue) blend films.

Figure S15, I am not sure if you can assign a slope to this “curves”

Reply: Thanks for the reviewer’s comments. We have added how the slope is assigned in

Figure S17 (order changed) note:

Figure S17. Energetics of non-fullerene based PBDTT-FTTE:O-IDTBR, PBDTT-FTTE:EH-IDTBR, PBDB-T:ITIC, PBDTT-FTTE:m-ITIC and PffBT4T-2OD:EH-IDTBR devices.

The charge carrier density as a function of voltage plots shows a slope at the range of 6.3 to 7.9 in these non-fullerene acceptor based device systems (the device thicknesses range between 70 to 115 nm). The slope γ is from the exponential fit $n = n_0 \exp(\gamma V_{oc})$ of the measured charge carrier density as a function of V_{oc} . The tail state slope E_{ch} derives via $E_{ch} = 1/(2\gamma)$ using the charge extraction results measured at open circuit.⁴

1. Rice, A. H. *et al.* Controlling vertical morphology within the active layer of organic photovoltaics using poly(3-hexylthiophene) nanowires and phenyl-C 61-butyric acid methyl ester. *ACS Nano* **5**, 3132–3140 (2011).
2. Feng, S. *et al.* Fused-Ring Acceptors with Asymmetric Side Chains for High-Performance Thick-Film Organic Solar Cells. *Adv. Mater.* **29**, 1–7 (2017).
3. Yang, Y. *et al.* Side-Chain Isomerization on an n-type Organic Semiconductor ITIC Acceptor Makes 11.77% High Efficiency Polymer Solar Cells. *J. Am. Chem. Soc.* **138**, 15011–15018 (2016).
4. Kirchartz, T. & Nelson, J. Meaning of reaction orders in polymer:fullerene solar cells. *Phys. Rev. B* **86**, 165201 (2012).
5. Ashraf, R. S. *et al.* The Influence of Polymer Purification on Photovoltaic Device Performance of a Series of Indacenodithiophene Donor Polymers. *Adv. Mater.* **25**, 2029–2034 (2013).
6. Kuwabara, J. *et al.* Direct Arylation Polycondensation: A Promising Method for the Synthesis of Highly Pure, High-Molecular-Weight Conjugated Polymers Needed for Improving the Performance of Organic Photovoltaics. *Adv. Funct. Mater.* **24**, 3226–

- 3233 (2014).
7. Kong, J. *et al.* Long-term stable polymer solar cells with significantly reduced burn-in loss. *Nat. Commun.* **5**, 5688 (2014).
 8. Lee, H. K. H. *et al.* Organic photovoltaic cells-promising indoor light harvesters for self-sustainable electronics. *J. Mater. Chem. A* **6**, 5618–5626 (2018).
 9. Shuttle, C. G., Hamilton, R., O'Regan, B. C., Nelson, J. & Durrant, J. R. Charge-density-based analysis of the current-voltage response of polythiophene/fullerene photovoltaic devices. *Proc. Natl. Acad. Sci.* **107**, 16448–16452 (2010).
 10. Shockley, W. & Read, W. T. Statistics of the Recombinations of Holes and Electrons. *Phys. Rev.* **87**, 835–842 (1952).
 11. Hall, R. N. Electron-Hole Recombination in Germanium. *Phys. Rev.* **87**, 387–387 (1952).

Reviewers' comments:

Reviewer #1 (Remarks to the Author):

I have read the manuscript "Tail state photocurrent collection..." by Wu and co-workers quite carefully now. The authors have added new data, new discussions, and clarifications which have greatly improved the article. However, while I find the work containing novel aspects that add to the topic of thick organic solar cell films, which is of much current interest, the discussions themselves still appear a bit convoluted, and at parts, confused. My concerns in the previous review appear to not have been fully addressed. Let me summarize my appreciation, again:

1. The central story is that for thick organic solar cell films, the key problem killing FF is the large voltage drop due to the space charge accumulating towards the distal exiting contact. This reduces the field inside the photogeneration region, which reduces the local carrier extraction efficiency. As a consequence, one needs sufficiently high carrier mobilities to avoid this problem, which can already be achieved in real materials today. Further if the photogeneration occurs closer to one contact, as it invariably will in thick films, then the carrier with the lower mobility should be arranged to leave the proximal contact rather than the distal contact. The essence of these features have already been explained in the literature in great detail as a consequence of the so-called "space-charge voltage penalty" in Ref 53 and 54. The authors should therefore clearly acknowledge these aspects in the introduction and discussions, and then show where their present work exceeds this earlier work. In my mind, there are at least few aspects to build this case: (i) confirmation that the space-charge voltage is indeed the dominant problem that causes internal-field redistribution that is detrimental to carrier sweepout from the photogeneration region, (ii) demonstration that this can be achieved in general in modern materials systems and in thicker devices than previously possible, (iii) the provision of detailed design rules, e.g., minimum mobility rules for proximal and distal carriers for the thicker films of current interest, and (iv) for a particular system DT-PDPP2T-TT), removal of trap states by purification facilitate the meeting of these requirements. All very nice.

2. Instead, the authors did their discussions in what appears to me along a tortuous path: (I) first discussing the "mobility-lifetime" product (and the newly added Hecht equation), (II) then showing that the devices do not behave with fully extended "depletion width"(Figs 2 and 3), (III) then showing that the limitation of these depletion widths is not high background doping, and (IV) finally, asserting this is due to exiting carriers trapped in tail states (Fig 5). The problem with this approach is that it is contradictory/confusing in a number of aspects: (I) The mobility-lifetime product has meaning only when photogeneration is more-or-less uniform across the photoactive layer, as in the case of Si cells and thin organic solar cells. Otherwise the carrier lifetime varies along the width of the device, as in the present case, where film width of interest extends towards micron lengthscale, while photogeneration occurs in the first 200 nm of the illuminated contact, limited by absorption coefficient. Therefore over a large portion of the film to the distal contact, there is no photogeneration and no recombination, only space charge density and its attendant electric field (as illustrated in Fig 6). So the discussion of lifetime-mobility products, and analysis of such effects, may not be the best way to think about this problem - they should be placed in Supplementary Information. The paper is already rather long. (II, III) The authors assume there is a depletion region, in analogy to standard inorganic p-i-n structures, then quantified this, and showed that it is really not the depletion width set by doping. This is rather circuitous, but not surprising. Most workers in the field already know that organics are not sufficiently doped without special efforts, and so they behave as more-or-less intrinsic materials. Incidentally, this consideration also suggests that the capacitance data collected in Fig S11 must have been done in ambient light, resulting in production of free carriers that limit the intrinsic width, which needs to be stated in the caption. (IV) Finally, the assertion that trap states are limiting photocurrent collection appears to be restrictive. What the authors have shown is the consequence of insufficient

mobility. Low mobility however can come from either large trap width/deep traps, and/or low mobility at the mobility edge. In other words, a materials system with low density of shallow trap states, but insufficient mobility at the mobility edge will also give exactly same problems. In fact, the authors seem to have found such a rare comparative study in BTR vs purified DT-PDPP2T-TT! Therefore discussions of trap energy is only one aspect of the problem, which should therefore be summarized in main text, but relegated also in full glory to Supplementary Information. The analysis of trap depth is indirect.

3. In contrast, it will be good to replace these with theoretical simulations of the drift-diffusion-recombination model, if possible, to clarify various limits of operation. For example, the exact bandbending in Fig 6 depends on mobility characteristics of the two carriers. This was drawn for one specific set of assumptions of symmetry between electrons and holes. What about the other possibilities? Higher or lower mobility, or imbalanced mobility/trap depth?

3. Finally, a small point. In Fig S12, something seems strange. Why are the electron-only devices becoming ohmic (J proportional to V) at high voltages, and the hole-only devices becoming non-ohmic (J proportional to V^4) at high voltages, in same materials system. In the absence of light, we expect both to tend towards V^2 in the absence of injection limitations. In the presence of light, we expect both to follow V^1 up to tens of mA/cm².

Therefore, may I recommend the authors to do a bold revision of their manuscript to bring out the enduring aspects of it. I think it is really nice work.

Reviewer #2 (Remarks to the Author):

The authors have carefully addressed all concerns raised by the reviewers. It is editor's turn to make decision.

Reviewer #3 (Remarks to the Author):

The authors have very carefully addressed all my comments and those of the other reviewers. This paper is a very detailed and comprehensive work and I recommend it for publication as it is.

Reviewer #1 (Remarks to the Author):

I have read the manuscript "Tail state photocurrent collection..." by Wu and co-workers quite carefully now. The authors have added new data, new discussions, and clarifications which have greatly improved the article. However, while I find the work containing novel aspects that add to the topic of thick organic solar cell films, which is of much current interest, the discussions themselves still appear a bit convoluted, and at parts, confused. My concerns in the previous review appear to not have been fully addressed. Let me summarize my appreciation, again:

1. The central story is that for thick organic solar cell films, the key problem killing FF is the large voltage drop due to the space charge accumulating towards the distal exiting contact. This reduces the field inside the photogeneration region, which reduces the local carrier extraction efficiency. As a consequence, one needs sufficiently high carrier mobilities to avoid this problem, which can already be achieved in real materials today. Further if the photogeneration occurs closer to one contact, as it invariably will in thick films, then the carrier with the lower mobility should be arranged to leave the proximal contact rather than the distal contact. The essence of these features have already been explained in the literature in great detail as a consequence of the so-called "space-charge voltage penalty" in Ref 53 and 54. The authors should therefore clearly acknowledge these aspects in the introduction and discussions, and then show where their present work exceeds this earlier work.

We have added a reference to the space charge penalty to the introduction, *page 5, line 101-105*. The focus of our paper is the effect of tail states on the quantitative amount of space charge built up and the specific discussion of the effect of space charge on the short-circuit current density. However we would also emphasize that whilst the referee's comment is focused on the effect of imbalanced mobilities (based on simulations) in limiting FF and PCE, our study is not focused on FF limitations, which we agree are largely understood, but rather focusing on factors limiting Jsc, which have received less attention.

In my mind, there are at least few aspects to build this case: (i) confirmation that the space-charge voltage is indeed the dominant problem that causes internal-field redistribution that is detrimental to carrier sweepout from the photogeneration region, (ii) demonstration that this can be achieved in general in modern materials systems and in thicker devices than previously possible, (iii) the provision of detailed design rules, e.g., minimum mobility rules for proximal and distal carriers for the thicker films of current interest, and (iv) for a particular system (DT-PDPP2T-TT), removal of trap states by purification facilitate the meeting of these requirements. All very nice.

2. Instead, the authors did their discussions in what appears to me along a tortuous path: (I) first discussing the "mobility-lifetime" product (and the newly added Hecht equation), (II) then showing that the devices do not behave with fully extended "depletion width" (Figs 2 and 3), (III) then showing that the limitation of these depletion widths is not high background doping, and (IV) finally, asserting this is due to exiting carriers trapped in tail states (Fig 5). The problem with this approach is that it is contradictory/confusing in a number of aspects: (I) The mobility-lifetime product has meaning only when photogeneration is more-or-less uniform across the photoactive layer, as in the case of Si cells and thin organic solar cells. Otherwise the carrier lifetime varies along the width of the device, as in the present case, where film width of interest extends towards micron lengthscale, while photogeneration occurs in the first 200 nm of the illuminated contact, limited by Absorption coefficient. Therefore over a large portion of the film to the distal contact, there is no photogeneration and no recombination, only space charge density and its attendant electric field (as

illustrated in Fig 6). So the discussion of lifetime-mobility products, and analysis of such effects, may not be the best way to think about this problem - they should be placed in Supplementary Information.

The referee is of course correct that mobility-lifetime product analysis has only limited validity. However, this analysis is widely used as a convenient figure of merit in many studies in the literature. We also find that even for this samples studies herein, this product determined from the thin device is a useful indicator of device FF for a wider range of device thickness, although not of J_{SC} , the focus of this study. Therefore we think it is appropriate to retain our discussion of this product, although to address the referee's justified concern, we have added a sentence in the introduction emphasising the limitations of mobility-lifetime product analyses at *page 4-5, line 94-101*.

The paper is already rather long. (II, III) The authors assume there is a depletion region, in analogy to standard inorganic p-i-n structures, then quantified this, and showed that it is really not the depletion width set by doping. This is rather circuitous, but not surprising. Most workers in the field already know that organics are not sufficiently doped without special efforts, and so they behave as more-or-less intrinsic materials.

We have added references to papers assigning space charge layer limitations to dark doping on *page 5, line 100 and page 12, line 273* to justify our inclusion of ruling out this possibility. For thin devices, we agree that it is accepted that dark doping is not important. However, there has been little discussion of how thick devices have to be before this effect becomes important.

Incidentally, this consideration also suggests that the capacitance data collected in Fig S11 must have been done in ambient light, resulting in production of free carriers that limit the intrinsic width, which needs to be stated in the caption.

The referee is correct on this point. We have added more detail on the measurement in the figure caption for this *figure S11*.

(IV) Finally, the assertion that trap states are limiting photocurrent collection appears to be restrictive. What the authors have shown is the consequence of insufficient mobility. Low mobility however can come from either large trap width/deep traps, and/or low mobility at the mobility edge. In other words, a materials system with low density of shallow trap states, but insufficient mobility at the mobility edge will also give exactly same problems. In fact, the authors seem to have found such a rare comparative study in BTR vs purified DT-PDPP2T-TT! Therefore discussions of trap energy is only one aspect of the problem, which should therefore be summarized in main text, but relegated also in full glory to Supplementary Information. The analysis of trap depth is indirect.

We agree with the referee's points on the potential importance of both band mobility and charge trapping. We have rephrased the text where we start to discuss tail states to indicate the wider importance of carrier mobilities, *page 15-16, line 360-369*. We have also restructured and shortened our discussion of tail states, effective mobilities and band mobilities, and directly linked this to our comparison of BTR and purified DT-PDPP2T-TT mobilities, as suggested by the reviewer – *page 16, lines 369-375*.

3. In contrast, it will be good to replace these with theoretical simulations of the drift-diffusion-recombination model, if possible, to clarify various limits of operation. For example, the exact

bandbending in Fig 6 depends on mobility characteristics of the two carriers. This was drawn for one specific set of assumptions of symmetry between electrons and holes. What about the other possibilities? Higher or lower mobility, or imbalanced mobility/trap depth?

The referee is correct that imbalanced mobilities can also impact on space charge layer effects, as previous studies referenced in our manuscript have reported. However as we don't have any direct experimental assays of such imbalances in the study we report herein, we don't think it would be helpful to include simulations of this effect, as these would be unlikely to go beyond previous literature studies. However, to clarify the position of the space charge layer and the role of tail states, we have added more explanation on *page 17-18, line 414-423*.

3. Finally, a small point. In Fig S12, something seems strange. Why are the electron-only devices becoming ohmic (J proportional to V) at high voltages, and the hole-only devices becoming non-ohmic (J proportional to V^4) at high voltages, in same materials system. In the absence of light, we expect both to tend towards V^2 in the absence of injection limitations. In the presence of light, we expect both to follow V^1 up to tens of mA/cm².

Our SCLC results were measured under dark condition. For the hole only device, in the low bias region, the current is about 3 orders of magnitude lower than the electron current in the electron only device, indicating ohmic behaviour. While, at higher voltage region, the abrupt increase of the current is suggested to be due to trap filling.¹ For the electron only device, the Mott-Gurney region appears at a relatively low voltage range, while at higher voltage, the current seems limited by the contacts.² It is very possible that in our simple device structure (now described in the *caption of Figure S12*), the contact barriers are not ideal, for this reason, these data are only included for indicative purposes. We have substantially expanded *the caption to Figure S12* to clarify these points.

1. Röhr, J. A., Moia, D., Haque, S. A., Kirchartz, T. & Nelson, J. Exploring the validity and limitations of the Mott–Gurney law for charge-carrier mobility determination of semiconducting thin-films. *J. Phys. Condens. Matter* **30**, 105901 (2018).
2. Blom, P. W. M., de Jong, M. J. M. & Vleggaar, J. J. M. Electron and hole transport in poly(p-phenylene vinylene) devices. *Appl. Phys. Lett.* **68**, 3308–3310 (1996).

Therefore, may I recommend the authors to do a bold revision of their manuscript to bring out the enduring aspects of it. I think it is really nice work.

REVIEWERS' COMMENTS:

Reviewer #1 (Remarks to the Author):

The authors have taken into consideration the issues raised, and greatly improved their manuscript.
Please publish.